# Global alignment and assessment of TRP channel transmembrane domain structures to explore functional mechanisms

Katherine E Huffer[1], Antoniya A Aleksandrova[2], Andrés Jara-Oseguera[1], Lucy R Forrest[2], Kenton J Swartz[1]*

[1]Molecular Physiology and Biophysics Section, Porter Neuroscience Research Center, National Institute of Neurological Diseases and Stroke, National Institutes of Health, Bethesda, United States; [2]Computational Structural Biology Section, Porter Neuroscience Research Center, National Institute of Neurological Diseases and Stroke, National Institutes of Health, Bethesda, United States

**Abstract** The recent proliferation of published TRP channel structures provides a foundation for understanding the diverse functional properties of this important family of ion channel proteins. To facilitate mechanistic investigations, we constructed a structure-based alignment of the transmembrane domains of 120 TRP channel structures. Comparison of structures determined in the absence or presence of activating stimuli reveals similar constrictions in the central ion permeation pathway near the intracellular end of the S6 helices, pointing to a conserved cytoplasmic gate and suggesting that most available structures represent non-conducting states. Comparison of the ion selectivity filters toward the extracellular end of the pore supports existing hypotheses for mechanisms of ion selectivity. Also conserved to varying extents are hot spots for interactions with hydrophobic ligands, lipids and ions, as well as discrete alterations in helix conformations. This analysis therefore provides a framework for investigating the structural basis of TRP channel gating mechanisms and pharmacology, and, despite the large number of structures included, reveals the need for additional structural data and for more functional studies to establish the mechanistic basis of TRP channel function.

**\*For correspondence:**
swartzk@ninds.nih.gov

## Introduction

Transient Receptor Potential (TRP) channels are a large and diverse family of cation permeable ion channel proteins that are expressed in animals and yeast, algae and other unicellular organisms. The biological functions of TRP channels are remarkably diverse, and include nociception, thermosensation, immune cell function, control of cellular excitability, fluid secretion, cardiac and smooth muscle function and development, ion homeostasis and lysosomal function (*Nilius and Flockerzi, 2014*; *Ramsey et al., 2006*; *Venkatachalam and Montell, 2007*). The family name is derived from the *Drosophila* mutant that causes blindness in which the neurons of mutant flies exhibit a transient receptor potential (trp) instead of a persistent response to illumination with intense light in electroretinograms (*Cosens and Manning, 1969*). The trp mutation was subsequently localized to the protein that functions as the phototransduction channel in the *Drosophila* retina (*Montell, 2011*). TRP channels have been classified into seven subfamilies: TRPC (canonical), TRPV (vanilloid), TRPM (melastatin), TRPA (ankyrin), TRPN (NOMPC), TRPP (polycystic) and TRPML (mucolipin) (*Clapham, 2007*). As expected from their widespread expression and physiological roles, mutations in TRP channels cause a range

of human diseases and are considered important drug targets for pain, inflammation, asthma, cancer, anxiety, cardiac disease and metabolic disorders (*Moran, 2018*; *Nilius et al., 2007*).

TRP channels have a notable historical significance in membrane protein structural biology because the structure of TRPV1 determined in 2013 ushered in a new era for solving near-atomic resolution structures of membrane proteins using cryo-electron microscopy (cryo-EM) (*Cao et al., 2013*; *Liao et al., 2013*). At least one structure has now been reported for each subfamily, with a total of 136 TRP channel structures available at the time we performed this analysis (*Autzen et al., 2018*; *Cao et al., 2013*; *Chen et al., 2017*; *Dang et al., 2019*; *Deng et al., 2018*; *Diver et al., 2019*; *Dosey et al., 2019*; *Duan et al., 2019*; *Duan et al., 2018b*; *Duan et al., 2018c*; *Fan et al., 2018*; *Grieben et al., 2017*; *Guo et al., 2017*; *Hirschi et al., 2017*; *Huang et al., 2018*; *Hughes et al., 2019*; *Hughes et al., 2018a*; *Hughes et al., 2018b*; *Hulse et al., 2018*; *Huynh et al., 2016*; *Jin et al., 2017*; *Liao et al., 2013*; *McGoldrick et al., 2019*; *McGoldrick et al., 2018*; *Paulsen et al., 2015*; *Saotome et al., 2016*; *Shen et al., 2016*; *Singh et al., 2019*; *Singh et al., 2018a*; *Singh et al., 2018b*; *Singh et al., 2018c*; *Su et al., 2018a*; *Su et al., 2018b*; *Tang et al., 2018*; *Vinayagam et al., 2018*; *Wang et al., 2018*; *Wilkes et al., 2017*; *Winkler et al., 2017*; *Yin et al., 2019a*; *Yin et al., 2019b*; *Yin et al., 2018*; *Zhang et al., 2018*; *Zheng et al., 2018*; *Zhou et al., 2017*; *Zubcevic et al., 2019a*; *Zubcevic et al., 2016*; *Zubcevic et al., 2018a*; *Zubcevic et al., 2018b*). These structures show that TRP channels are tetramers, with each subunit containing six transmembrane (TM) helices (S1-S6), and with the S5 and S6 helices from the four subunits forming a central pore domain containing the ion permeation pathway (*Figure 1*). The S1-S4 helices form peripheral domains within the membrane with a domain-swapped architecture such that each S1-S4 domain is positioned near to the pore-forming S5-S6 helices from the adjacent subunit (*Figure 1A,B*). The N- and C-termini contribute to forming large intracellular domains that differ extensively between subfamilies (*Figure 1F–M*). Most TRP channels also contain a highly conserved helical extension of the pore-lining S6 helix named the TRP box that projects through a tunnel formed by the intracellular-facing surface of the S1-S4 domain and the pre-S1 region of the N-terminus (*Figure 1F,I–M*). In many instances, structures of the same TRP channel have been determined in the absence and presence of activating ligands and toxins, inhibitors, or with mutations that promote open or closed states, providing a wealth of information about the structural basis of their functional properties and pharmacology.

To synthesize what has been learned from these TRP channel structures, and to provide a framework for comparing structural elements in functionally critical regions, we generated a structure-based alignment of the transmembrane domains for most of the available TRP channel structures. We used the structural alignment to compare key regions of the ion permeation pathways in the context of their roles in ion selectivity and gating, as well as binding sites for ligands and regulatory ions. Remarkably, even though our analysis considers an unprecedented number of related ion channel structures, it identified the need for additional structural data and for more functional studies to establish the mechanistic basis of TRP channel function and pharmacology.

## Results

### Structure-based alignment of TRP channels

Sequence-based alignment of TRP channels is complicated by low sequence identity, with a previous multiple sequence alignment of TRP channel TM sequences revealing just 16% identity as the major mode of the full multiple sequence alignment (*Palovcak et al., 2015*). Structure-based alignments are thought to be more reliable than sequence-based alignments, particularly when sequence identity is low (*Carpentier and Chomilier, 2019*). In addition, structure-based alignments are sensitive to conformational changes and can reveal how residues may change position during ligand binding, channel opening, or other conformational changes. To interrogate relationships of functionally important regions within the TM domains, therefore, we aimed to generate a structure-based alignment for all available TRP channels. The availability of a large number of TRP channel structures allows for a more comprehensive structure-based alignment than has previously been performed for this or other protein families, and we believe that similar structural alignments would provide useful perspective for other protein families with low sequence homology and many available structures.

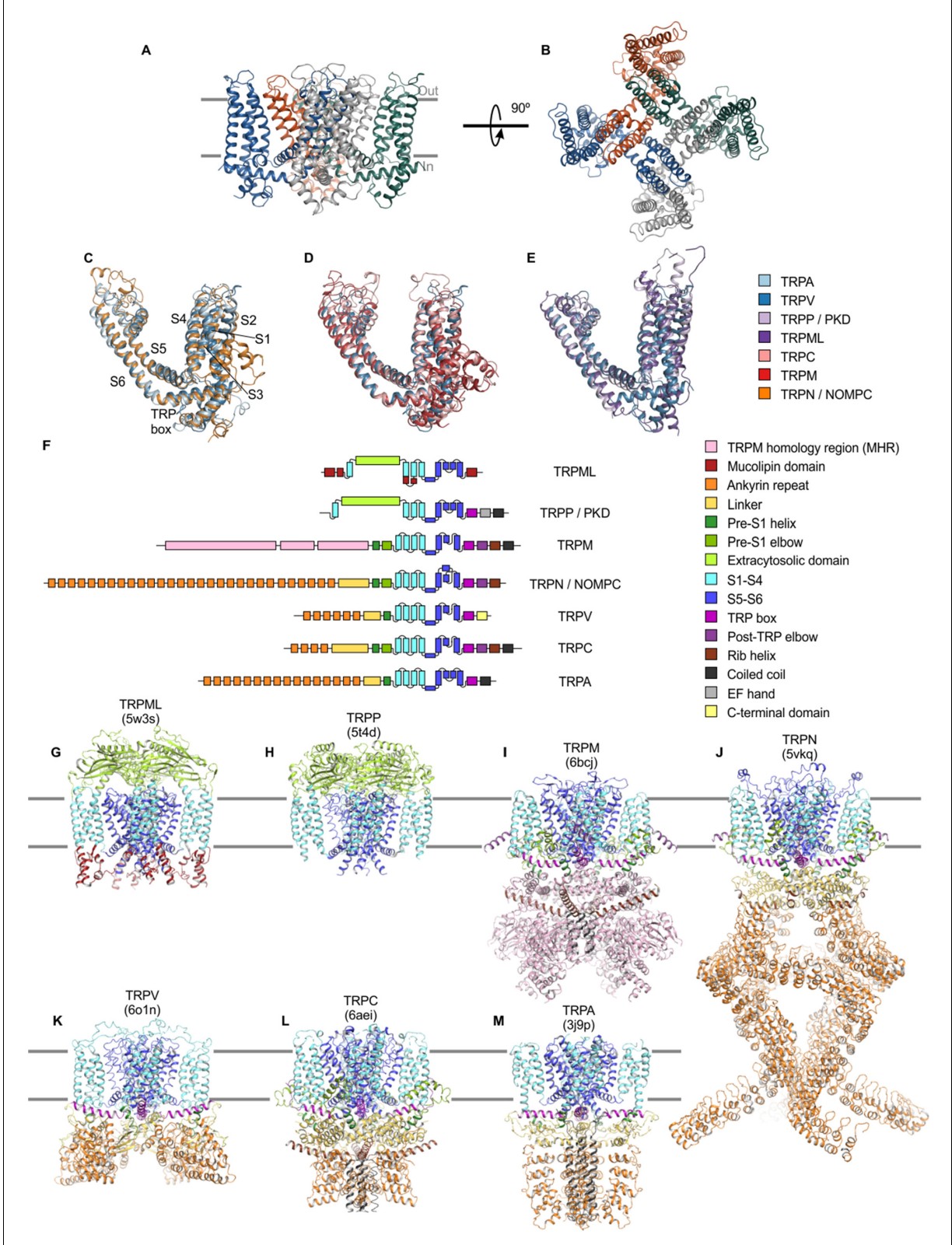

**Figure 1.** Structures of TRP channel subfamilies. (A) Backbone fold of apo TRPV1 in nanodiscs (5irz) viewed from the side, with approximate boundaries of the membrane indicated with gray bars. (B) Same structure as in A viewed from the extracellular side. (C–E) Superimposed structures of TM domains for apo TRPV1 (blue; 5irz) with (C) apo TRPN (orange; 5vkq) and apo TRPA1 (light blue; 3j9p), (D) apo TRPC5 (pink; 6aei) and apo TRPM4 (red; 6bcj). (E) apo TRPML3 (dark purple; 5w3s) and apo TRPP1 (light purple; 5t4d). For clarity, only one of the four subunits shown. (F) Schematic of domain

*Figure 1 continued on next page*

*Figure 1 continued*

architecture of TRP channel subunits. (**G–M**) Cartoon representation of structure with domains colored as in F of apo structures of (**G**) TRPML3 (5w3s), (**H**) TRPP1 (5t4d), (**I**) TRPM4 (6bcj), (**J**) TRPN (5vkq), (**K**) TRPV5 (6o1n), (**L**) TRPC5 (6aei), and (**M**) TRPA1 (3j9p).

A total of 136 TRP channel structures had been reported at the time of this analysis, of which 117 were determined using cryo-EM and 22 using X-ray diffraction. A subset of these structures, however, are of limited resolution and their inclusion would have complicated the analysis. We therefore selected 120 of those structures that were resolved to effective resolutions of 5 Å or better (***Figure 2—source data 1***) and focused on the TM region, which appears to have a well-conserved architecture across TRP channels. It is important to note that although our quality control cutoff was defined based on the nominal overall resolution of the experimental electron density maps, the analyses in this study were performed on the models built to fit that data rather than the electron density itself. Differences in model quality arising from, for example, local resolution of maps, goodness of model fit to maps, and quality of model geometry will affect the accuracy of our analysis.

The TM regions of these channels were aligned using Fr-TM-Align (***Pandit and Skolnick, 2008***; ***Zhang and Skolnick, 2005***), which aligns structures pairwise by optimizing for the global template-modeling-score (TM-score), a measure of backbone fold similarity that is independent of protein length (see Materials and methods). As a fragment-based alignment method, Fr-TM-Align is effective even in cases with large conformational differences (***Stamm and Forrest, 2015***). Alignments of the TM regions of the TRP family structures generally have TM-scores of >0.6, indicating that they share similar global folds (***Figure 2***; ***Xu and Zhang, 2010***). The aligned TRP channel structures also share a common fold with a voltage-activated potassium channel (2r9r; TM-scores ranging from 0.46 to 0.78, with TM-scores <0.6 obtained only for some TRPM and TRPC structures), consistent with them sharing six TM helices per subunit, a common tetrameric assembly and a domain-swapped architecture. As negative controls, we compared the TRP channels to two structurally unrelated channels (trimeric P2X3 and pentameric ELIC) (***Mansoor et al., 2016***; ***Pan et al., 2012***) and obtained TM-scores ranging from 0.08 to 0.40, consistent with the mean TM-score of 0.3 obtained for the best alignments between randomly selected proteins (***Zhang and Skolnick, 2004***).

The cytoplasmic domains of TRP channels adopt unique folds between subfamilies and thus have been traditionally used to define subfamilies. Nevertheless, sequence analysis of the TM regions alone is sufficient to define TRP channel subfamilies (***Palovcak et al., 2015***; ***Yu and Catterall, 2004***). To evaluate the quality of our structural alignments, we examined whether segregation into subfamilies could be observed using hierarchical clustering based on TM-score alone (***Figure 2***). With a few notable exceptions, clustering based on TM-score corresponded nicely to existing subfamily assignments, despite the variety of methods of structure determination (X-ray vs cryo-EM) and imaging environments (detergent, amphipol, or nanodisc) used, suggesting that the conditions of structure determination have not introduced substantial artifacts (***Figure 2***). Where possible, we have directly compared structures of the same complex determined using cryo-EM and X-ray crystallography and observed high TM-scores and close association in the hierarchical clustering, indicating that structures determined by different methods are indeed similar (e.g. apo rTRPV6, TM-score = 0.96 for 6bob and 5wo7; vanilloid agonist-bound TRPV2 quadruple mutant, TM-score = 0.88 for 6o07 and 6bwj; apo TRPV2, TM-score = 0.87 for 5an8 and 6bwm). Note that, because the TM-score is normalized by the length of the reference protein, the TM-scores for a given pair of proteins are asymmetric depending on which protein is chosen as the reference. As noted in the Materials and methods, we chose to perform hierarchical clustering along the stationary protein axis, so that the TM-scores compared were for different mobile proteins to the same stationary protein.

On the sequence level, the pore domains in TRP channels are highly conserved across all TRP channel subfamilies, whereas the peripheral S1-S4 domains are more variable between subfamilies (***Figure 2—figure supplement 1***; sequence identity data from Fr-TM-Align pairwise alignments not shown) (***Ng et al., 2019***; ***Palovcak et al., 2015***; ***Vinayagam et al., 2018***). Consistent with this pattern, clustering of TRP channel structures into subfamilies was more robust when considering the TM-scores of the peripheral S1-S4 domains compared to those of the pore domain (***Figure 2—figure supplements 2*** and ***3***).

In the hierarchical clustering of the entire TM region, the structures determined for *Xenopus laevis* TRPV4 (6bbj) and the rTRPV6 L495Q mutant (5iwk) are notable exceptions because the TM-scores

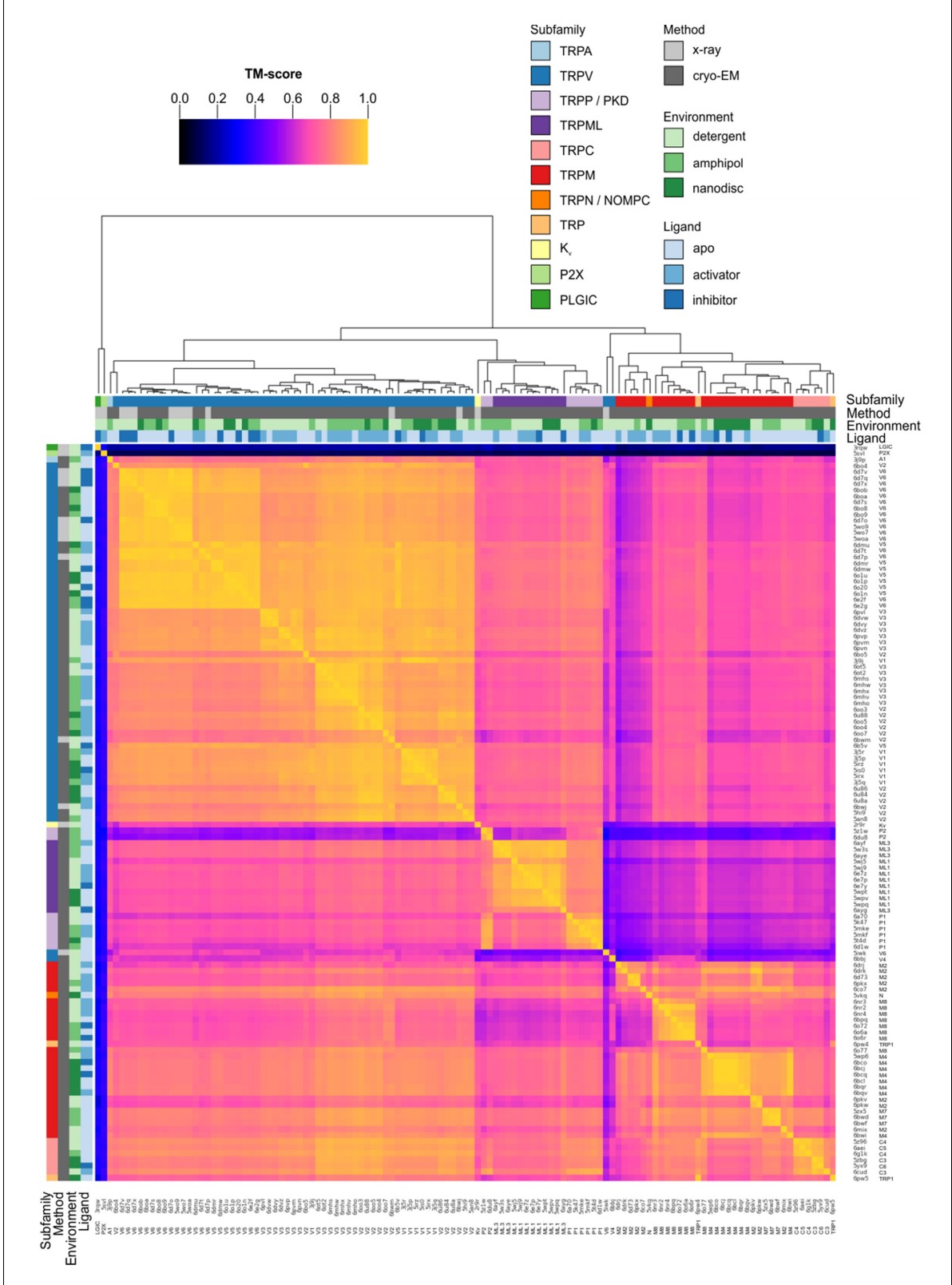

**Figure 2.** Clustered heatmap of TM-scores for the pairwise structural alignments of the TM domains of TRP channels. The heatmap indicates how similar the TM domain (pre-S1 to TRP box) of each pair of TRP channel structures is, as expressed by the TM-score reported for each pairwise structural alignment performed with Fr-TM-align. A comparison of identical structures is indicated with yellow (TM-score of 1), whereas no structural similarity is indicated with black (TM-score of 0). The TM-score is further used to cluster the TRP channel structures. Representative Kv, P2X and PLGIC channel

*Figure 2 continued on next page*

*Figure 2 continued*

structures are included as a control. Note that the heatmap is not symmetric because the TM-score takes into account the sequence length of the reference structure (see Materials and methods).

The online version of this article includes the following source data and figure supplement(s) for figure 2:

**Source data 1.** Master TRP channel list.
**Source data 2.** Data file for clustered heatmap of TM domain.
**Source data 3.** Data file for clustered heatmap of pore domain.
**Source data 4.** Data file for clustered heatmap of S1-S4 domain.
**Figure supplement 1.** Structure-based multiple sequence alignment for TRP channels structures.
**Figure supplement 2.** Clustered heatmap of TM-scores for the pairwise structural alignments of the pore domains of TRP channels.
**Figure supplement 3.** Clustered heatmap of TM-scores for the pairwise structural alignments of the S1-S4 domains of TRP channels.
**Figure supplement 4.** Multiple sequence alignment for TRP channel sequences whose structures have been determined and used for structure-based alignment.

---

of their alignments with the other TRPV channels were unusually low (0.41 to 0.77 for TRPV4 and 0.32 to 0.73 for rTRPV6 L495Q) (*Figure 2*). When the pore domain and S1-S4 domains were considered separately, the rTRPV6 L495Q mutant did cluster with the vanilloid subfamily (*Figure 2—figure supplements 2* and *3*), consistent with the individual domains adopting similar folds and with the global fold dissimilarity in the whole TM region stemming from the rTRPV6 L495Q mutant adopting a non-domain-swapped architecture unlike the wild-type protein (*Saotome et al., 2016*; *Singh et al., 2017*). In the case of TRPV4 (*Deng et al., 2018*), the S1-S4 and pore domains did not cluster with other TRPV channels even when those domains were considered independently (*Figure 2—figure supplements 2* and *3*), but the reason for this structural distinction is not clear. It will therefore be important to determine additional structures of the TRPV4 channel to determine whether the distinct architecture in the TM region is defining for this subtype. The P2X3 and ELIC channels that served as negative controls were not closely associated with any TRP channels after hierarchical clustering (*Figure 2*). From this analysis, we conclude that our structural alignment of TRP channels is robust and consequently that the backbone folds within the TM regions of these channels are most similar within subfamilies.

To enable comparison of structurally equivalent residues between TRP channel structures, we used the pairwise alignments of all 119 structures relative to the reference structure of *Nematostella vectensis* TRPM2 (nvTRPM2, 6co7) to construct a multiple sequence alignment (see Materials and methods; *Figure 2—figure supplement 1*). nvTRPM2 was selected as the reference as it contains the longest sequence in the TM regions, which served to maximize the length of the alignment. When compared to a sequence-based alignment of the same TM domain sequences generated using Clustal-Omega (*Madeira et al., 2019*) (see Materials and methods; *Figure 2—figure supplement 4*), the structure-based alignment identified relationships more accurately between proteins within all six TM helices, even in regions where sequence similarity is low, consistent with previous findings (*Carpentier and Chomilier, 2019*). Therefore, the structure-based alignment has been used in subsequent analysis. Relationships are more ambiguous in the reentrant pore loop that forms the ion selectivity filter near the extracellular end of the pore, reflecting considerable structural differences between subfamilies. In addition, the pre-S1 helix found in TRPM channels is unique to that subfamily and thus was not aligned to other TRP channels in this region.

## Dimensions of the intracellular S6 gate region in TRP channels

Regulation of ion flow across the membrane is a critical function of TRP channels, so investigation of ion permeation pathways in TRP channel structures is of obvious interest. Two regions that are critical for ion permeation are the external ion selectivity filter and the internal S6 activation gate. With respect to the latter, functional studies examining the accessibility of introduced cysteine residues to thiol-reactive compounds and metals (*Salazar et al., 2009*), as well as studies examining the state-dependence of blocking ions (*Jara-Oseguera et al., 2008*; *Oseguera et al., 2007*), have concluded that the TRPV1 channel opens and closes at the intracellular end of the pore in response to vanilloid binding. A similar S6 activation gate region has been identified in studies of structures determined in the absence of activating ligands for all TRP channel subfamilies, with the exception of a few that have high baseline activity, such as TRPV5 and TRPV6 (*Chen et al., 2017*; *Guo et al., 2017*;

*Hirschi et al., 2017*; *Jin et al., 2017*; *Liao et al., 2013*; *Paulsen et al., 2015*; *Schmiege et al., 2017*; *Shen et al., 2016*; *Tang et al., 2018*; *Winkler et al., 2017*). Therefore, the intracellular end of the pore is of key interest when examining the conduction pathway of TRP channels.

To globally assess ion permeation pathways in TRP channels, we calculated the accessibility of those pathways using HOLE (*Smart et al., 1996*), after further restricting our analysis to structures for which side-chains for all pore-lining residues have been assigned (see Materials and methods; *Figure 3*; *Figure 3—figure supplements 1* and *2*). We also identified those residues responsible for determining the dimensions of the ion permeation pathway and mapped minimum radius values onto the structure-based sequence alignment for S6 and for those elements contributing to the ion selectivity filter (See Materials and methods; *Figure 4*). In all structures, the intracellular S6 constrictions occur at one or more of four positions spanning three helical turns of the S6 helix, suggesting that depending on the S6 helix conformation, a cytosolic gate could be formed at different sites (*Figure 3*; *Figure 3—figure supplements 1* and *2*; *Figure 4*). The deepest of these constrictions within the pore we designated as site A and the one closest to the cytoplasmic surface as site D, with sites B and C being the most common locations of the narrowest S6 constriction across TRP channel subfamilies (*Figure 4*).

When considering the ability of ions to permeate, we took into account the structural characteristics of the pores, in particular the hydrophobicity and the afforded diameter of the conduction pathway. At a constriction where polar side chains or backbone carbonyls can contribute to ion coordination, such as the extracellular selectivity filter of TRP channels, ions may pass through in a partially or fully dehydrated state, with a lower bound ionic radius of approximately 1 Å for fully dehydrated $Na^+$ or $Ca^{2+}$ ions. At a hydrophobic constriction such as the one formed by the intracellular S6 helices, hydrophobic side chains will not attract ions or facilitate ion dehydration, meaning that ions likely pass the S6 gate in a fully hydrated state with effective radii of >3 Å for hydrated $Na^+$, $K^+$, $Ca^{2+}$, and $Mg^{2+}$ ions (*Nightingale, 1959*).

For all available TRP channel structures, the open probability of the construct used for structure determination has not been measured in either the absence or presence of activating stimuli, hindering objective attempts to relate specific structures to distinct functional states. If we consider only the 55 TRP channel structures with no ligands modeled in the structure as representing apo states (see Materials and methods, *Figure 2—source data 1*), 38 contain multiple regions along the intracellular side of the S6 helix at which the pore radius is ≤1.0 Å, too narrow to support permeation of hydrated cations (*Figure 3*; *Figure 3—figure supplements 1* and *2*; *Figure 4*), even considering the inherent dynamics of the structure. Dehydrated cations are also unlikely to permeate given the hydrophobic nature of the contributing side chains at the S6 constrictions. Notably, there are several examples of apo state channel structures in which the pore radius near the intracellular end of the S6 helices is wider than 1.0 Å (*Figure 3*; *Figure 3—figure supplements 1* and *2*; *Figure 4*). Specifically, in the case of mouse and human TRPV3 and rabbit TRPV5, the pore radius within the S6 gate can be as large as 2 Å, whereas for rabbit TRPV2 and human TRPV6 (*Figure 3—figure supplement 1* and *2*), as well as for TRPM2, TRPP1 (PKD2) and TRPP2 (PKD2L1) the minimal pore radius can be as large as 3 Å. Although some of these S6 gate regions are nearly large enough to allow permeation of hydrated monovalent and divalent permeant cations (radii from 3.3 Å for $Na^+$ to 4.1 Å for $Ca^{2+}$), for each of these subtypes other apo structures have been determined with internal pores narrower than a radius of 1.2 Å (*Figure 3*; *Figure 3—figure supplements 1* and *2*; *Figure 4*). Given that the internal pores in all TRP channels are lined by hydrophobic residues (*Figure 4*; *Figure 4—figure supplement 1*), and thus would not attract ions nor facilitate ion dehydration, it seems likely that most of the TRP channel structures discussed thus far represent non-conducting states where the S6 gate is closed. TRPV6 is interesting because this channel has a relatively high open probability in cellular membranes (0.25–0.9 depending on voltage and the concentration of phosphatidylinositol 4,5-bisphosphate, abbreviated $PIP_2$) (*Zakharian et al., 2011*), and therefore one would expect the structures would be more likely to correspond to open conformations than for other TRP channels. In fact, many of the TRPV6 structures contain S6 gates narrower than 1.2 Å and thus likely represent closed conformations. However, in all these cases, the protein used for structure determination contained truncations or mutations that might have influenced the closed-open equilibrium. In contrast, two structures of wild-type human TRPV6 and one of the Y467A mutant contain S6 gates with minimal radii of 2.7–3.1 Å, suggesting that they may represent an open, ion-conducting state (*McGoldrick et al., 2018*). Nevertheless, it is unclear whether opening of the S6 gate to this extent

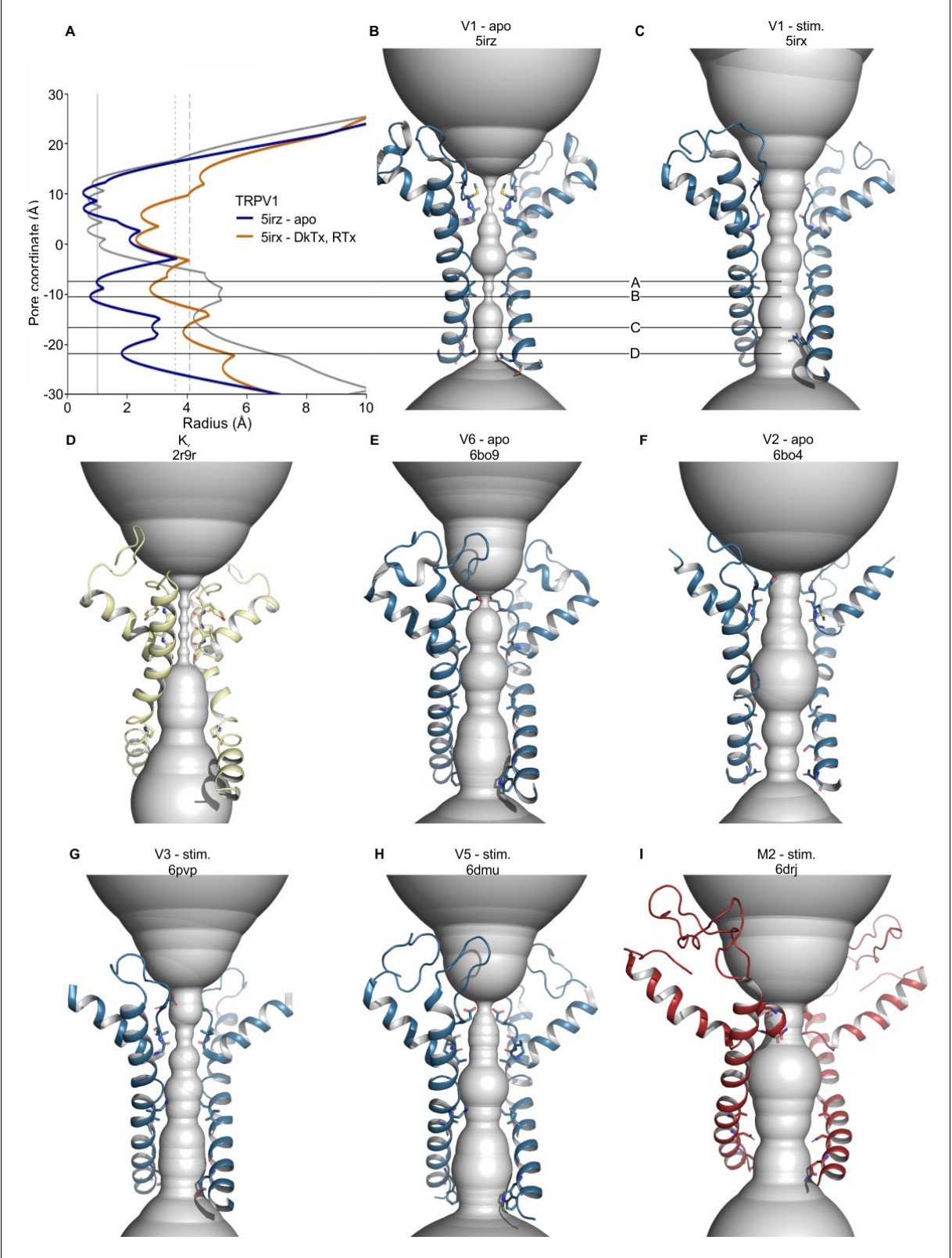

**Figure 3.** Pore radii in the S5-S6 pore domains of selected TRP channels according to HOLE. (A) Pore radius profiles for TRPV1 apo (blue, 5irz) and RTx/DkTx complex (orange, 5irx) structures and for the Kv1.2/2.1 paddle chimera (grey, 2r9r). Vertical lines indicate radii of dehydrated $Na^+$ and $Ca^{2+}$ ions (solid), hydrated $Na^+$ (dotted, $^HNa^+$) and $Ca^{2+}$ ions (dashed, $^HCa^{2+}$). (B–I) Backbones of tetrameric cation channel pore domains, focusing on structures with the widest S6 gate radii, with HOLE representations in gray for (B) apo TRPV1 (5irz), (C) TRPV1 with RTx/DkTx bound (5irx), (D) Kv 1.2/2.1 paddle

*Figure 3 continued on next page*

*Figure 3 continued*

chimera (2r9r), (**E**) apo TRPV6 (6bo9), (**F**) apo TRPV2 (6bo4), (**G**) TRPV3 Y564A mutant after pretreatment at 37°C (6pvp), (**H**) TRPV5 with $PIP_2$ (6dmu), and (**I**) TRPM2 with $Ca^{2+}$ and ADP-ribose (6drj).

The online version of this article includes the following source data and figure supplement(s) for figure 3:

**Source data 1.** Data file containing HOLE profile analysis for TRP channel pore domains.
**Source data 2.** Data file for minimum SF and S6 radii scatterplot.
**Figure supplement 1.** HOLE pore radius representations for S5-S6 pore domains of representative TRP and Kv channels.
**Figure supplement 2.** HOLE pore radius representations for S5-S6 pore domains of representative TRP channels.
**Figure supplement 3.** Comparing selectivity filter and internal pore minimum radii for TRP channel structures.

can support a single channel conductance of 30–50 pS, as measured for human TRPV6 (*Zakharian et al., 2011*).

When considering those structures determined in the presence of activators (*Figure 2—source data 1*), it is notable that 16 out of 35 contain pores narrower than 1.0 Å radius in the cytoplasmic region, suggesting that they represent non-conducting (possibly desensitized) states (*Figure 3*; *Figure 3—figure supplements 1*, *2*, *3*; *Figure 4*). Only 19 of these activator-bound structures have more dilated internal pores, with radii ranging from 1.2 to 4.4 Å. Of all the TRP channel structures reported thus far, that of zebrafish TRPM2 bound to its two activators (ADP-ribose and $Ca^{2+}$, 6drj) is the most likely to represent an open state, as the dimensions of the S6 gate region of zebrafish TRPM2 (4.4 Å radius) (*Huang et al., 2018*) are similar to those of Kv channel structures widely considered to be open (4.2–15 Å radius) (*Hite and MacKinnon, 2017*; *Long et al., 2007*; *Tao et al., 2017*; *Tao and MacKinnon, 2019b*; *Wang and MacKinnon, 2017*; *Figure 3*; *Figure 3—figure supplement 1*). The state of the remaining activator-bound structures is more ambiguous, as dimensions of the S6 gate region range from radii of 3 Å for rat TRPV1 and mouse and human TRPV3 to 3.3 Å for rabbit TRPV5 (3j5q, 6pvp and 6dmu, respectively). In the case of TRPV1, not only is the single channel conductance quite high (90–100 pS at positive voltages) (*Hui et al., 2003*; *Oseguera et al., 2007*; *Premkumar et al., 2002*), but also quaternary ammonium blocking ions as large as tetrapentyl ammonium (10 Å diameter) must be able to pass the S6 gate when open (*Jara-Oseguera et al., 2008*; *Oseguera et al., 2007*), suggesting that the cytoplasmic pore is likely to be larger than a minimal radius of 3 Å. In addition, although the open probability of the construct of TRPV1 used for structure determination is not known, it contains a deletion of the pore-turret that is known to decrease open probability below 0.5 (*Geron et al., 2018*; *Jara-Oseguera et al., 2016*).

Surprisingly, there was no striking correlation between the dimensions of the internal pore and whether the protein structure was determined in the absence or presence of activators or inhibitors (*Figure 3—figure supplement 3*). The prevalence of a cytoplasmic constriction across TRP channel subfamilies supports the prevailing idea that the internal region of S6 functions as a universal gate, and it seems likely that in most instances the structure of a fully open state remains to be determined.

## The ion selectivity filter in TRP channels

The extracellular end of the ion permeation pathway is relatively narrow in most structures and across TRP channel subfamilies (*Figure 3*; *Figure 3—figure supplements 1* and *2*), consistent with this region serving as an ion selectivity filter as it does in related tetrameric cation channels (*Owsianik et al., 2006*). Notably, in several instances such as apo flycatcher TRPM8, the structure of the selectivity filter is poorly resolved, resulting in large pore radii in our analysis (*Yin et al., 2019a*; *Yin et al., 2018*). However, the external end of the pore is better resolved in a recent structure of great tit TRPM8 in the presence of activators (6o77) (*Diver et al., 2019*), suggesting that this region forms an ion selectivity filter similar to that in other TRP channels, albeit with dimensions that are less narrow (*Figure 3—figure supplement 1*).

The ion selectivity of TRP channels fits into three broad categories: $Ca^{2+}$ selective (TRPV5 and TRPV6), monovalent cation selective (TRPM4 and TRPM5), and non-selective among cations (all other TRP channels) (*Owsianik et al., 2006*). To assess whether there is any clear structural correlate to these differences in ion selectivity, we examined the available structures and identified three structural features of the ion selectivity filter that are consistently discernible for those channels that

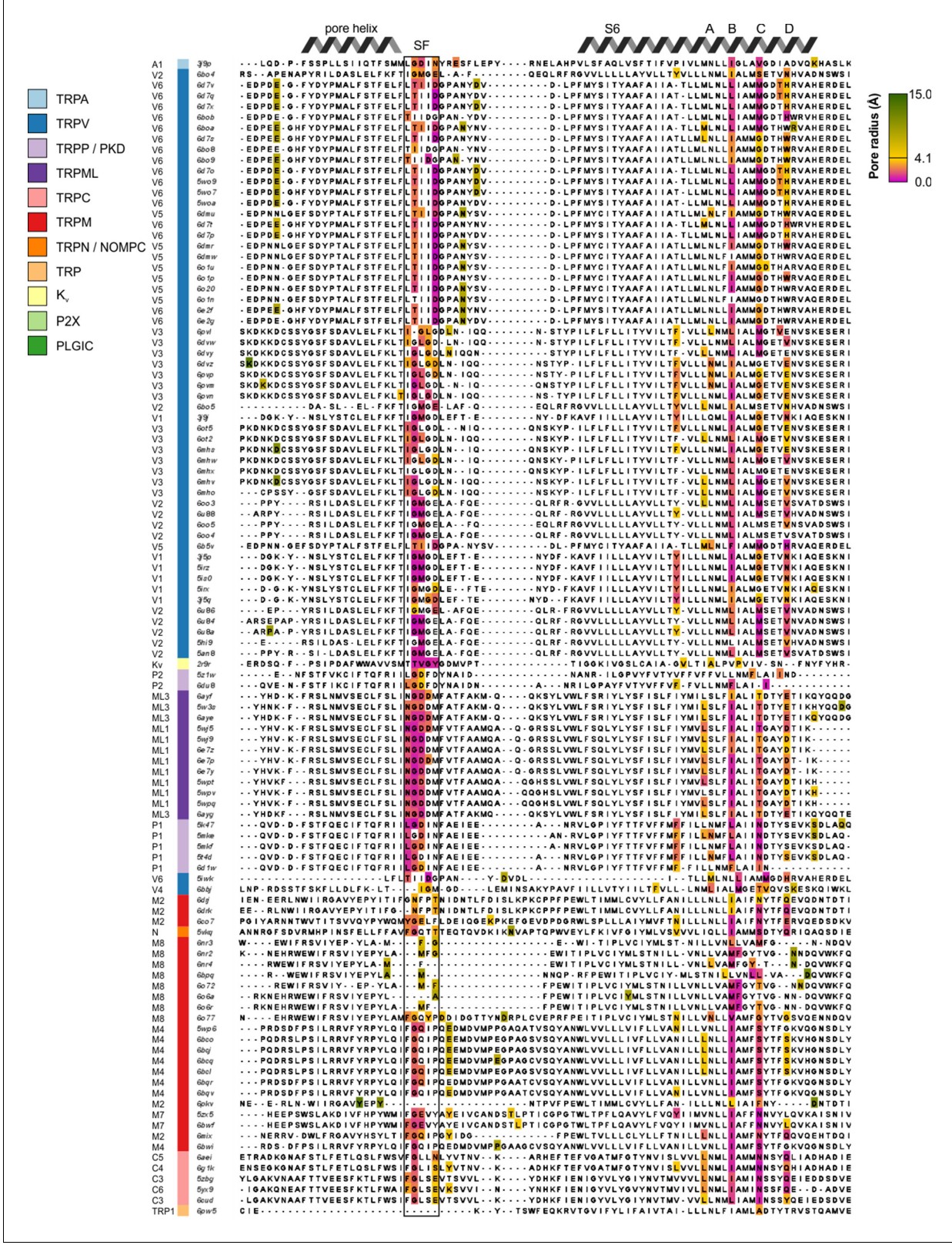

**Figure 4.** Structure-based multiple sequence alignment of pore-lining residues. Structure-based multiple sequence alignment of pore-lining residues, with pore-contributing residues colored based on the narrowest pore radius associated with any atom in that residue (from magenta for narrowest to green for widest, passing through gold at 4.1 Å to represent the radius of a hydrated $Ca^{2+}$ ion). Uncolored residues do not have any atoms whose van

*Figure 4 continued on next page*

*Figure 4 continued*

der Waals radii intersect with the HOLE profile. Sequences are ordered based on hierarchical clustering from *Figure 2*. The selectivity filter is indicated by SF.

The online version of this article includes the following source data and figure supplement(s) for figure 4:

**Source data 1.** Excel file.
**Figure supplement 1.** Structure-based multiple sequence alignment of pore-lining residues.

---

are non-selective between monovalent and divalent cations, some of which were noted in the original report of the structure of human TRPP1 based on a comparison to structures of TRPV1, TRPV2 and TRPA1 (*Shen et al., 2016*; *Figure 5*; *Figure 4—figure supplement 1*). First, a highly conserved Gly residue (G643 in TRPV1) is present in non-selective channels after the C-terminal end of the reentrant pore helix that enables a sharp bend in the backbone of the protein (*Figure 5*; *Figure 4—figure supplement 1*). Second, two backbone carbonyls are positioned towards the base of the filter where they could interact with permeant ions (*Figure 5*; *Figure 4—figure supplement 1*). The presence of backbone carbonyls within a narrow region of the filter is reminiscent of $K^+$ channels where ion dehydration is critical for ion permeation and selectivity (*Doyle et al., 1998*; *Zhou et al., 2001*). Third, the side chain of an acidic residue or a Gln is positioned immediately adjacent to the extracellular side of the narrowest region of the filter (*Figure 5*; *Figure 4—figure supplement 1*). TRPV1-4 channels also contain a second conserved Gly residue within the narrowest region of the filter, which positions the conserved Asp or Gln within the permeation pathway (*Figure 4—figure supplement 1*). Mutagenesis of the conserved Asp in TRPV1 and TRPV4 channels reduces divalent ion permeability and channel affinity for the inhibitor ruthenium red (*García-Martínez et al., 2000*; *Voets et al., 2002*). Although these features are consistently seen in all structures of non-selective TRP channels, the dimensions of the filter vary considerably (with minimal radii from 0.5 to 3.7 Å for structures in which the selectivity filter is resolved) (*Figure 3—figure supplements 1* and *2*; *Figure 4*), raising the possibility that the filters of these TRP channels have intrinsic flexibility. These conserved features of the selectivity filter arise despite low sequence identity or similarity between nonselective TRP channels from different subfamilies (*Figure 5—figure supplements 1* and *2*).

Notably, all three of these structural features seen in non-selective TRP channels are discernably different in the two $Ca^{2+}$-selective channels, TRPV5 and TRPV6, as originally noted for the X-ray structure of TRPV6 (*Saotome et al., 2016*). In place of the conserved Gly after the reentrant pore helix, TRPV5 and TRPV6 contain a conserved Thr residue that contributes its hydroxyl group to the ion permeation pathway (*Figure 5*; *Figure 4—figure supplement 1*). In addition, these $Ca^{2+}$-selective channels have a more extended selectivity filter that contains at least three backbone carbonyl groups positioned to line the permeation pathway (*Figure 5*; *Figure 4—figure supplement 1*). The conserved Asp or Gln residues found in the non-selective cation permeable TRP channels is always an Asp in TRPV5 and TRPV6, and mutagenesis of this Asp is known to diminish $Ca^{2+}$ permeation and $Mg^{2+}$ block in TRPV5 (*Nilius et al., 2001*), Finally, the pore radius is consistently narrower at this region of the filter for those two $Ca^{2+}$ selective channels (*Figure 4*; *Figure 5*; *Figure 4—figure supplement 1*); indeed, density attributable to a divalent ion was identified in this external narrow region in the X-ray structure of rat TRPV6 (*Saotome et al., 2016*).

The structures of the monovalent cation-selective TRPM4 channel from human and mouse are intriguing because these channels exhibit most of the key features seen in the structures of non-selective TRP channels, including the conserved first Gly and two backbone carbonyls within the permeation pathway, and the conserved acidic/Gln position at the external end of the filter is always a Gln (*Figure 5*; *Figure 4—figure supplement 1*). The side chain of this Gln was noted to hydrogen bond with the backbone carbonyl of the conserved first Gly in adjacent subunits in one TRPM4 structure, and this network was proposed to stabilize the filter with a diameter large enough to support permeation of hydrated monovalent ions, but not large enough for hydrated divalent ions nor narrow enough to permit ion coordination and dehydration (*Autzen et al., 2018*; *Duan et al., 2018c*; *Guo et al., 2017*; *Winkler et al., 2017*). Mutation of the conserved Gln to Glu, Asp or Asn disrupts the monovalent cation selectivity of the TRPM4 channel (*Guo et al., 2017*; *Nilius et al., 2005*), indicating a critical role of this residue and supporting the proposed mechanism of monovalent cation selectivity. However, it is noteworthy that the dimensions of the filter vary from 1.4 to 2.3 Å in the

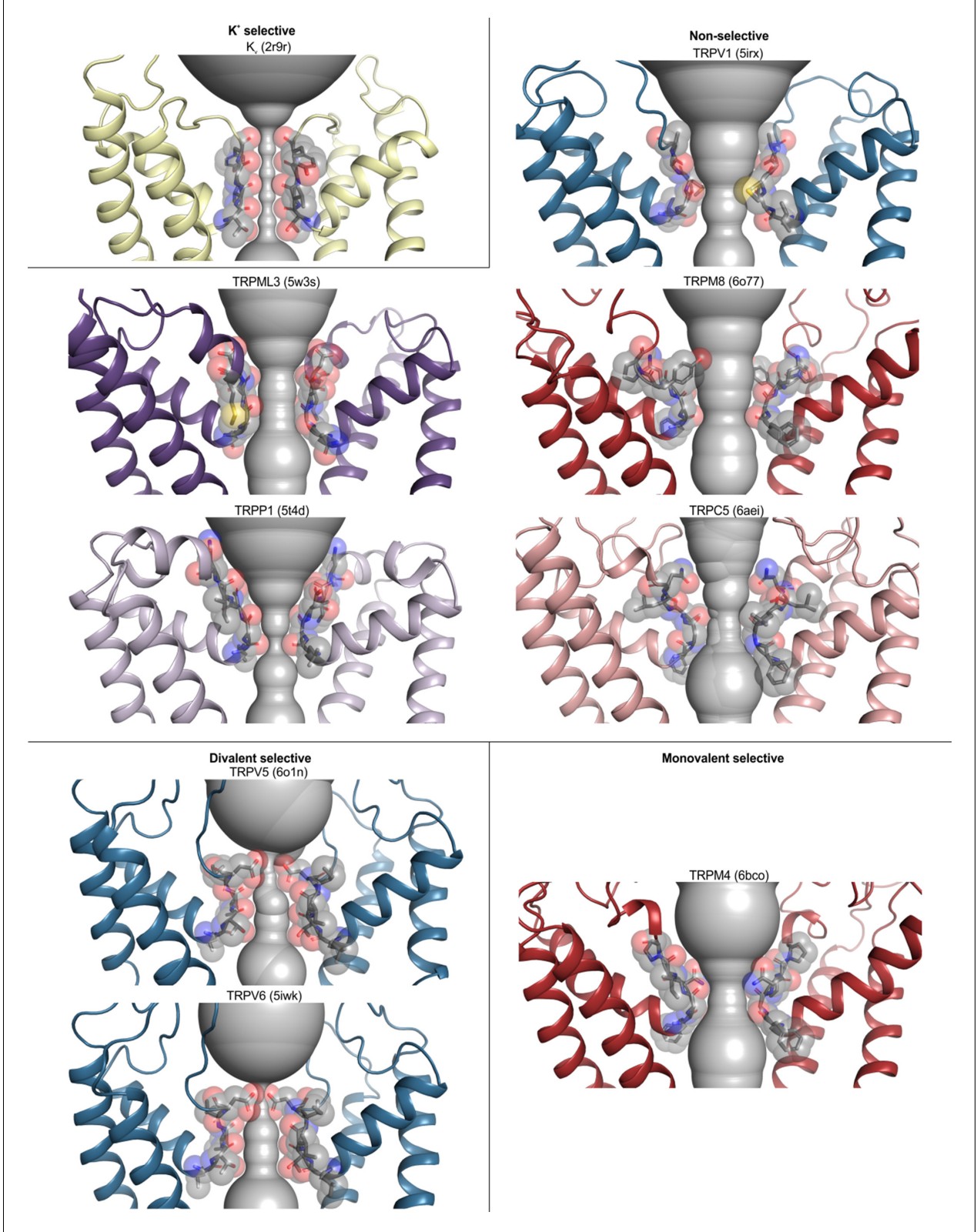

**Figure 5.** Selectivity filters in TRP channels with different ion selectivity. Pore domains (S5–S6) of selected TRP channels, focusing on the selectivity filter constriction at the extracellular end of the pore. Residues lining the selectivity filter are shown as grey sticks and spheres, with HOLE profiles shown in grey. PDB accession codes are 2r9r (Kv 1.2/2.1 paddle chimera), 5irx (TRPV1), 5w3s (TRPML3), 6o77 (TRPM8), 5t4d (TRPP1), 6aei (TRPC5), 6o1n (TRPV5), 5iwk (TRPV6), and 6bco (TRPM4). While TRPM5 is also classified as monovalent-selective, there are no structures available for this channel.

*Figure 5 continued on next page*

*Figure 5 continued*

The online version of this article includes the following figure supplement(s) for figure 5:

**Figure supplement 1.** Heatmap comparing selectivity filter identity in TRP channels.

**Figure supplement 2.** Heatmap comparing selectivity filter similarity in TRP channels.

available TRPM4 structures (Figure 3—Source data 1) and no structures have yet been reported for TRPM5 channels, the only other TRP channel that exhibits monovalent cation selectivity. It is interesting that a subset of non-selective channels (e.g. TRPM2 and TRPM8) also contain a Gln at this position, indicating that this residue is not sufficient to determine monovalent cation selectivity and that the dimensions of the pore and the water coordination geometry are likely critical to the mechanism of ion selectivity. Indeed, the ion selectivity filter of great tit TRPM8 recently resolved in a desensitized state (6o77) (*Diver et al., 2019*) reveals that the dimensions of the filter are considerably larger than seen in monovalent-selective TRPM4 structures (*Autzen et al., 2018*; *Duan et al., 2018c*; *Guo et al., 2017*; *Winkler et al., 2017*). A particularly important feature to resolve going forward will be to determine whether ions permeate a given selectivity filter in hydrated or partially dehydrated forms. A reasonable working hypothesis emerging from these structures is that monovalent-selective channels may largely conduct hydrated cations, divalent cation-selective channels may largely conduct dehydrated cations, while non-selective channels may permit both hydrated and dehydrated forms of cations to permeate. However, a thorough mechanistic understanding of ion permeation in TRP channels will require additional experimental and computational studies to determine the energetic contributions from pore flexibility and nearby charges.

In addition to the heterogeneity in dimensions of the selectivity filters noted above, in several TRP channel structures, the ion selectivity filters clearly adopt distinct conformations in apo state structures compared to those that have activators bound. For example, in the case of TRPV1, the filter has a minimum radius of ~0.5 Å in the apo structure but expands to a minimum radius of 2.5 Å in the presence of the activating toxins double-knot toxin and resiniferatoxin (DkTx and RTx, respectively; *Figure 3*), leading to the proposal that the selectivity filter in TRPV1 might also serve as a gate that regulates ion permeation (*Cao et al., 2013*; *Gao et al., 2016b*). The idea of two gates has been extended to other TRP channels in which structural rearrangements in the ion selectivity filter are discernable, including TRPV2 (*Huynh et al., 2016*; *Zubcevic et al., 2016*; *Zubcevic et al., 2019b*; *Zubcevic et al., 2018b*) and TRPP2 (PKD2L1) (*Grieben et al., 2017*; *Shen et al., 2016*; *Su et al., 2018b*; *Wilkes et al., 2017*). A correlate of this proposal is that the apo form would not conduct ions, as the narrow dimensions of the selectivity filters in TRPV1 and TRPV2 appear incompatible with hydrated ion passage. In contrast, selectivity filters with minimal radii <1.0 Å allow ion permeation in K$^+$ channels, where ion dehydration is thought to be central to the mechanism of ion selectivity (*Doyle et al., 1998*; *Zhou et al., 2001*; *Figure 3*; *Figure 3—figure supplement 2*). Despite their similar selectivity filter dimensions, TRPV1 and TRPV2 channels, unlike K$^+$ channels, contain a hydrophobic methionine in the selectivity filter that would prevent ion dehydration and thus ion permeation in solved conformations. Recent functional experiments examining the possible role of the selectivity filter as a gate in TRPV1-3 channels revealed that thiol-reactive Ag$^+$ ions permeated the selectivity filters in the absence of activators, suggesting that the filters of these channels allow ion permeation in the closed state (*Jara-Oseguera et al., 2019*). This study also demonstrated state-dependent changes in the accessibility of larger thiol reactive compounds, supporting the idea that the filter changes conformation during channel activation. Further investigation of the physiologically accessible conformations and dynamics of TRP channel selectivity filters is required to understand the functional significance of conformational dynamics of their filters.

## Ligand-binding pockets in the TM domains of TRP channels

TRP channels are activated by a diverse array of chemical ligands and stimuli such as temperature (*Clapham, 2007*), yet the structures of the TM regions to which many of these activators bind are remarkably similar. Although vanilloid sensitivity has been engineered into both TRPV2 and TRPV3 (*Zhang et al., 2016*; *Zhang et al., 2019*), suggesting that the gating mechanisms of these vanilloid-insensitive TRP channels are similar to those of TRPV1, we currently understand very little else about how the gating mechanisms of different TRP channel subfamilies are related. Out of the 120

available TRP channel structures that we analyzed, 30 were determined in complex with activating ligands contacting the TM region, including vanilloids (e.g. RTx; TRPV1 and TRPV2), DkTx (TRPV1), cooling agents (icilin and WS-12; TRPM8), $Ca^{2+}$ ions (TRPM2, TRPM4 and TRPM8), cannabidiol (CBD; TRPV2), ML-SA1 (TRPML1), 2-aminoethoxydiphenyl borate (2-APB; TRPV3 and TRPV6) and PIP$_2$ (TRPM8) (*Figure 2—source data 1*). These structures provide an unprecedented opportunity to explore the structure and conservation of ligand-binding sites across different TRP channels. Densities for interacting lipids can also be seen in the maps for many TRP channel structures, but we omitted these from our analysis because in most cases the quality of the cryo-EM density is insufficient to unambiguously identify the lipid. To explore the extent to which ligand-binding sites are conserved between different TRP channels, for each ligand, we selected a template structure in complex with that ligand, identified any residues with side chain atoms within 4 Å of the ligand and then used our structure-based sequence alignments to examine the corresponding residues in all other structures. For each ligand, we defined a sequence motif representing all residues lining the ligand binding pocket regardless of their location along the primary sequence of the channel, calculated the percentage of identical and similar residues in the corresponding motif in all other structures, and generated corresponding heat maps and structure-based sequence alignments (*Figure 6*; *Figure 6—figure supplements 1* and *2*; see Materials and methods).

The vanilloid-binding pocket observed in the rat TRPV1-RTx complex in nanodiscs (*Gao et al., 2016b*), and in an engineered rabbit TRPV2 channel in detergent (*Zubcevic et al., 2018b*), is positioned at the interface between the S1-S4 domain of each subunit and the pore-forming S5-S6 domain of the adjacent subunit, with residues in S3, S4, S4-S5 linker, S5 and S6 contacting RTx (*Figure 7A,B*). RTx is a relatively large ligand, with a surface area of 1,605 Å$^2$, and it contacts the side chains of 15 aliphatic and aromatic hydrophobic residues and four polar or charged residues in the complex with TRPV1 (*Figure 7B*). Notably, lipid-facing cavities lined by hydrophobic residues resembling this vanilloid-binding pocket in TRPV1 and TRPV2 can be seen in all TRP channel subfamilies (*Figure 6*; *Figure 6—figure supplement 1*). The similar side-chain character of residues lining the vanilloid-binding pocket in other TRP channels suggests that other hydrophobic ligands might bind to this pocket and raises the possibility that engineering vanilloid sensitivity into other TRP channel subfamilies, as has already been done for TRPV2 and TRPV3 (*Zhang et al., 2016*; *Zhang et al., 2019*), might be an informative approach to explore the extent to which gating mechanisms have been conserved. Indeed, the vanilloids capsaicin and capsazepine have been reported to inhibit TRPM8 channels (*Behrendt et al., 2004*; *Weil et al., 2005*), possibly by binding to the equivalent pocket, although their site and mechanism of action in the TRPM8 channel have yet to be explored. Density and mutagenesis also indicate that the TRPC6 inhibitor BTDM binds to an analogous location in TRPC6 (EMD-6856) (*Tang et al., 2018*).

DkTx contains two domains, K1 and K2, which bind to the outer perimeter of the pore domain of TRPV1, interacting with residues in the extracellular end of S6 and the reentrant pore helix, as well as with lipids in the surrounding membrane (*Figure 7*; *Bae et al., 2016*; *Gao et al., 2016b*). Interactions of the toxin with the channel involve a larger surface area than for the other ligands that activate TRP channels, as DkTx has a total surface area of 6377 Å$^2$, with protein-protein interfaces of 655 Å$^2$ and 556 Å$^2$ for the K1 and K2 domains, respectively (*Bae et al., 2016*), and involve both hydrophobic and polar interactions. MD simulations of the toxin-channel complex suggest that Y631, F649, T650, N652, D654, F655, K656, A657 and V658 on rTRPV1 interact with DkTx (*Bae et al., 2016*). Of these, mutations at Y631, F649, T650 and A657 are known to alter activation of the channel by the toxin (*Bohlen et al., 2010*). Our analysis of a more recent structure in nanodiscs further identifies K535, S629, S632, L635, I660, and I661 as being within 4 Å of DkTx (*Gao et al., 2016b*). DkTx is thought to be selective for TRPV1 as the toxin does not activate TRPV2, TRPV3, TRPV4, TRPA1 or TRPM8 (*Bohlen et al., 2010*). Although the residues in TRPV1 that likely interact with DkTx are not well conserved among other TRP channels (*Figure 6*; *Figure 6—figure supplement 1*), the interaction of DkTx with the surrounding lipid membrane is thought to be energetically important for binding (*Bae et al., 2016*; *Sarkar et al., 2018*) and could conceivably facilitate binding of the toxin to other TRP channels. Thus, attempting to engineer DkTx-sensitivity into other TRP channels might be a useful approach for exploring the extent to which gating mechanisms are conserved, in particular for channels where conformational changes in the external pore play important roles in gating.

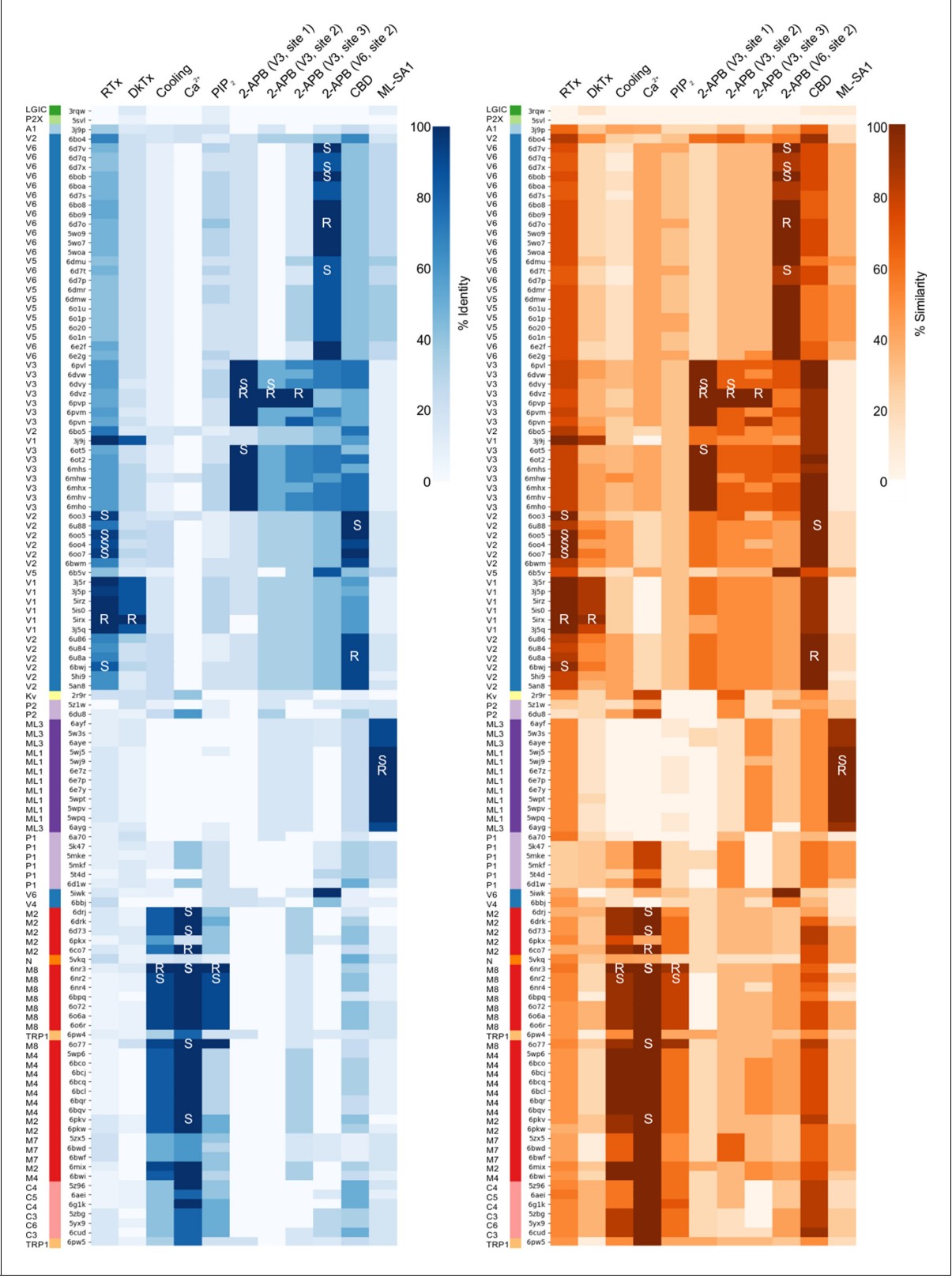

**Figure 6.** Heatmaps comparing ligand binding pocket motifs in TRP channels. Ligand-binding motifs were identified as including any residues with a side chain within 4 Å of the ligand. Heatmaps comparing the ligand-binding motifs in all structures based on percent identity (0–100, white to blue) or similarity (0–100, white to orange) when compared to the reference ligand-binding motif indicated with the letter R. Additional structures in which the

*Figure 6 continued on next page*

*Figure 6 continued*

ligand is also found are indicated with S (for secondary). Ligand-protein interactions are shown in *Figure 7*, *Figure 8*, *Figure 9*, and *Figure 9—figure supplement 1*. Sequences are ordered based on hierarchical clustering from *Figure 2*. Color code for TRP channels is from *Figure 2*.

The online version of this article includes the following source data and figure supplement(s) for figure 6:

**Source data 1.** Ligand motif identity heatmap data.
**Source data 2.** Ligand motif similarity heatmap data.
**Figure supplement 1.** Multiple sequence alignments for ligand-binding pocket motifs in TRP channels.
**Figure supplement 2.** Multiple sequence alignments for ligand binding pocket motifs in TRP channels.

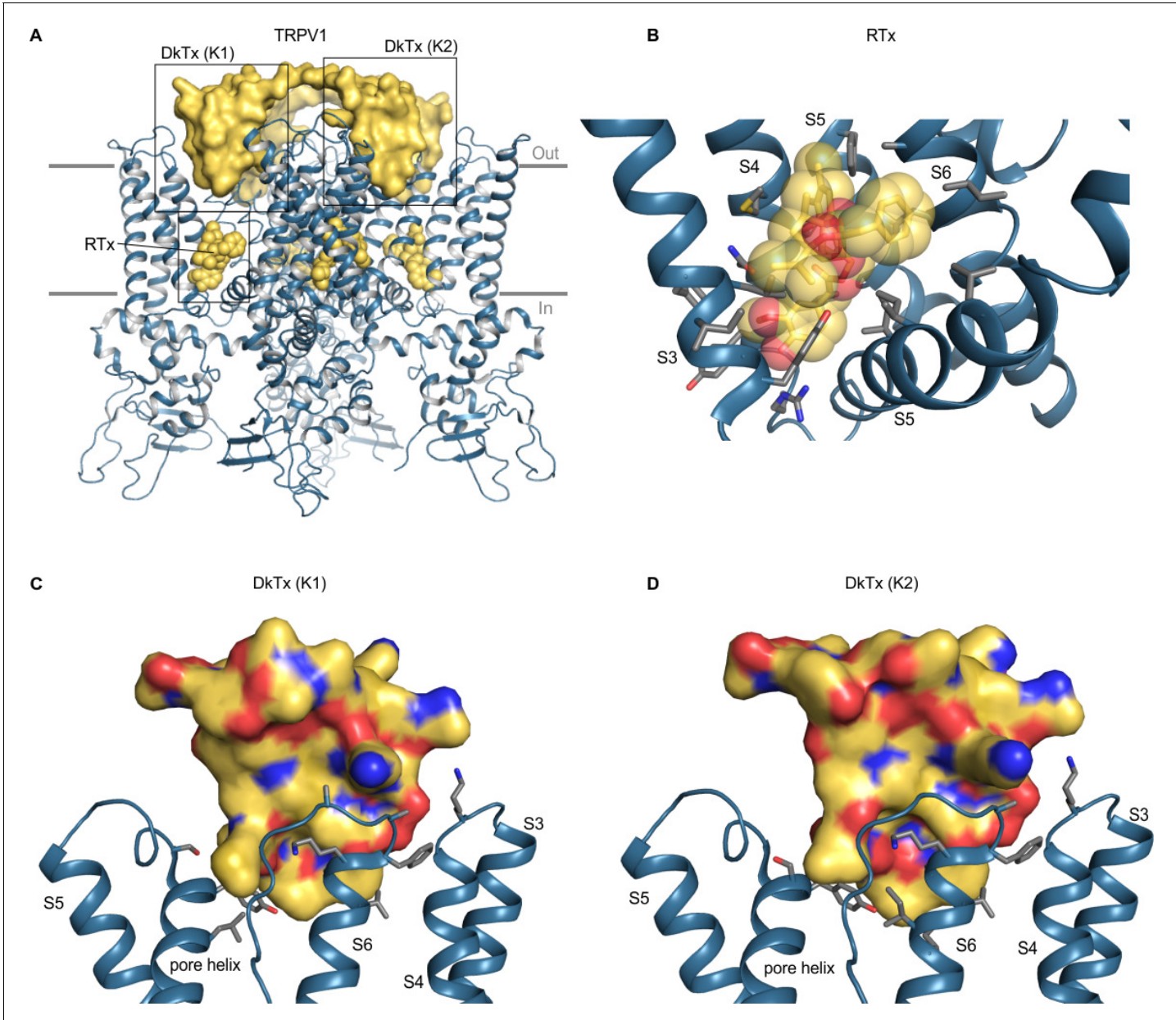

**Figure 7.** Structure of TRPV1 with RTx and DkTx bound. (**A**) Structure of TRPV1 in nanodiscs with RTx and DkTx shown as yellow spheres and yellow surface, respectively (5irx). (**B**) Close-up view of the RTx binding pocket with side chains colored by atom: carbon (gray), oxygen (red) and nitrogen (blue). (**C,D**) Close-up views of the DkTx binding surface showing either K1 or K2 knots, with linker omitted and side chain coloring as in B. Views are from the central pore axis looking out toward the lipid membrane. For clarity, helices without binding pocket residues have been hidden in panels B-D.

Cryo-EM structures of flycatcher TRPM8, in complex either with WS-12, a potent analog of menthol, or with both the cooling agent icilin and $Ca^{2+}$ ions, reveal that the cooling agent binding pocket is located close to the vanilloid-binding pocket seen in TRPV1 and TRPV2 (*Yin et al., 2019a*; *Yin et al., 2018*). In contrast to the vanilloid site, however, the cooling-agent-binding pocket is located entirely within the S1-S4 domain, with residues in all four helices and the TRP box contributing to the site (*Figure 8*). This cooling agent site is exposed to the surrounding membrane between the S2 and S3 helices (partially occluded by an α helix C-terminal to the TRP box), but also to the intracellular aqueous environment, which presumably allows $Ca^{2+}$ ions to access the site from the cytoplasm. The cooling-agent-binding pocket is considerably smaller than the vanilloid-binding pocket (the surface areas of icilin and WS-12 are 788 $Å^2$ and 742 $Å^2$, respectively) and contains many polar residues, with two Arg, two Tyr and one His residue positioned with atoms within 4 Å of the ligands (*Figure 8*). In addition, in the $Ca^{2+}$-icilin complex the intracellular end of the S4 helix adopts an alternate conformation that repositions residues in the binding pocket, a difference that is not seen for WS-12 (*Yin et al., 2019a*). These differences, along with differing $PIP_2$ engagement between the channel complexes with WS-12 or icilin, suggest that different cooling agents might have distinct mechanisms of activation or that the structures captured by WS-12 and icilin represent different physiological states. Beyond TRPM8, our analysis shows that residues lining the cooling-agent-binding pocket in TRPM8 are highly conserved in TRPM2 and TRPM4 structures and somewhat conserved in TRPM7 channels, but are very different in TRPV, TRPML and TRPP channels (*Figure 6*; *Figure 6—figure supplement 1*). Although it is unclear whether it would be possible to engineer cooling-agent-binding sites into other TRP channels given the relatively small size of the cavity and involvement of polar residues, it would be interesting to investigate why TRPM2 and TRPM4 have not been reported to be sensitive to cooling agents.

Intracellular $Ca^{2+}$ regulates the activity of TRPM2, TRPM4 and TRPM8 channels and densities attributed to $Ca^{2+}$ ions have been identified within the S1-S4 domains of all three TRPM channels (*Autzen et al., 2018*; *Diver et al., 2019*; *Huang et al., 2019*; *Huang et al., 2018*; *Yin et al., 2019a*; *Yin et al., 2018*; *Zhang et al., 2018*; *Zhao et al., 2020*). The $Ca^{2+}$ binding sites identified in these TRPM channels involve Glu, Asp, Gln and Asn residues in S2 and S3, and, in the context of TRPM8, the $Ca^{2+}$ binding site is contiguous with the cooling-agent-binding pocket, though none of the $Ca^{2+}$-binding residues directly contact the cooling agents WS-12 or icilin (*Figure 8B*). Our analysis shows that the $Ca^{2+}$-coordinating residues are not conserved in TRPV, TRPML or TRPP channels, nor in the more closely related TRPM7 channel, similar to the trend observed for the cooling-agent-binding pocket (*Figure 6*; *Figure 6—figure supplement 1*). Interestingly, the $Ca^{2+}$ binding motif is somewhat conserved in TRPC channels (*Figure 6*; *Figure 6—figure supplement 1*). Moreover, in cryo-EM structures of mouse TRPC4 and TRPC5 (5z96 and 6aei) densities were identified in the same site, although these densities were tentatively attributed to $Na^+$ ions based on buffer composition (*Duan et al., 2019*; *Duan et al., 2018a*). TRPC channels have been implicated in intracellular $Ca^{2+}$ signaling (*Curcic et al., 2019*), but whether $Ca^{2+}$ ions bind directly to the channel and regulate activity remains unclear. It would be interesting to mutate the putative ion binding site in TRPC channels to explore whether $Ca^{2+}$ can directly modulate channel activity through this ion binding site. Our structure-based alignment does not reveal notable residue similarity to the $Ca^{2+}$ ion binding site in TRPA1 (*Figure 6*). However, recent structural and functional work on TRPA1 has demonstrated an electron density and similar contributing side chains consistent with a $Ca^{2+}$ binding site, and mutations in this site disrupt TRPA1 modulation by $Ca^{2+}$ (*Zhao et al., 2020*). These inconsistencies appear to be due to structural differences between the included chimeric TRPA1 structure (3j9p) and the recently solved $Ca^{2+}$-bound structures of human TRPA1 (6v9v and 6v9w).

The 2-APB-binding sites identified in cryo-EM structures of mouse and human TRPV3 and rat and human TRPV6 channels are noteworthy because this ligand functions as either an activator or inhibitor for many different TRP channels and was observed at three distinct sites, designated sites 1–3 (*Figure 9*; *Singh et al., 2018a*; *Singh et al., 2018c*; *Zubcevic et al., 2019a*). Site one in mouse and human TRPV3 is located within the cytoplasm at the interface between the TRP helix and the pre-S1 helix (*Figure 9C*) and mutations in this site also alter the apparent affinity for 2-APB (*Hu et al., 2009*; *Singh et al., 2018a*; *Zubcevic et al., 2019a*). Site two in mouse TRPV3 is located near the intracellular end of the TM domains between the S1-S4 domain and the TRP helix (*Figure 9D*), in the vicinity of the cooling-agent-binding sites in TRPM8 (*Singh et al., 2018a*). This site is similar to that identified for rat and human TRPV6 using X-ray crystallography, and the ligand density was

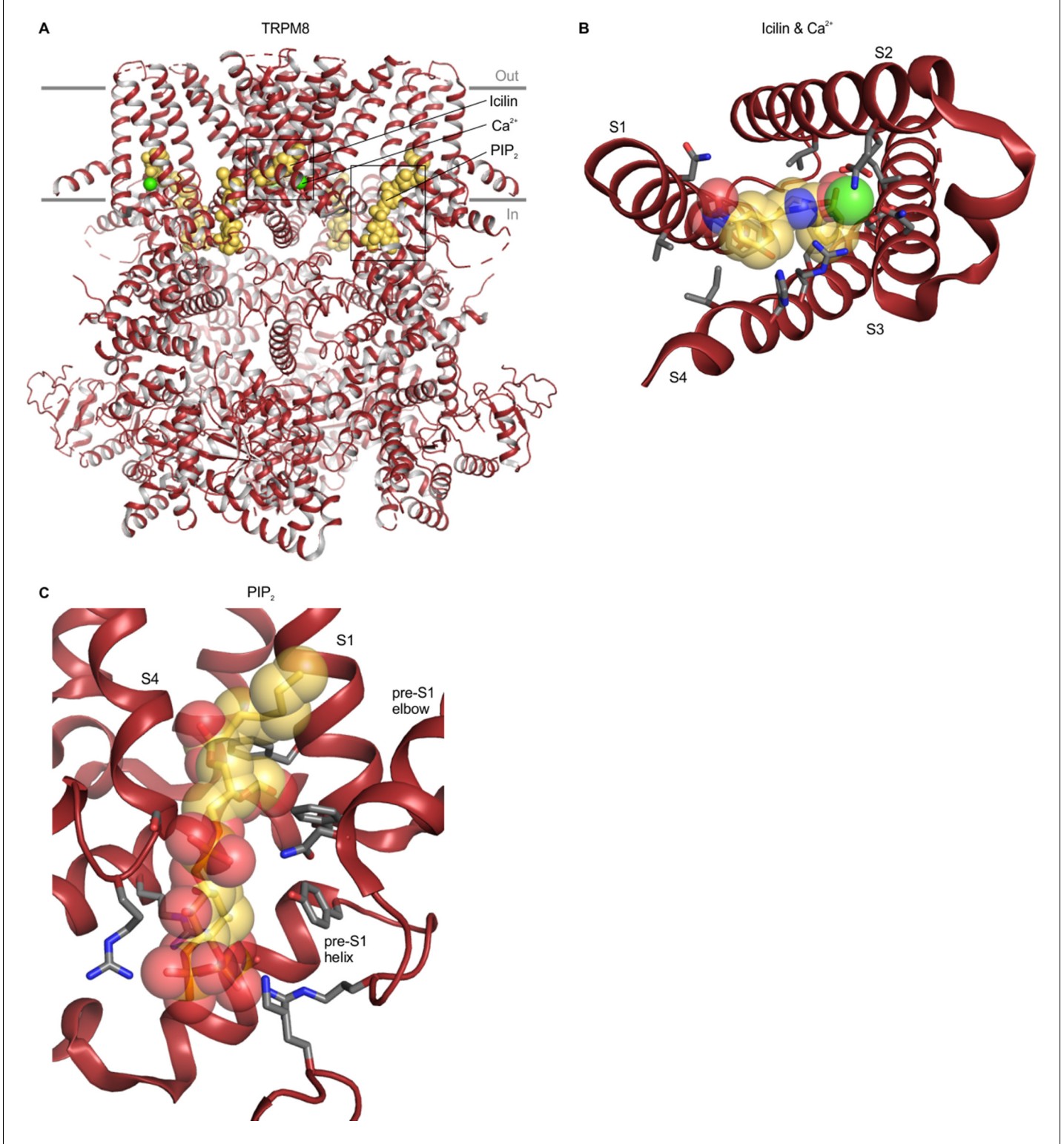

**Figure 8.** Structure of TRPM8 with icilin, $Ca^{2+}$ and $PIP_2$ bound. (**A**) Structure of TRPM8 with icilin, $Ca^{2+}$ and $PIP_2$ bound (6nr3), with yellow spheres for ligands and green spheres for $Ca^{2+}$. (**B**) Close-up view of the icilin and $Ca^{2+}$ binding site from the intracellular side of the membrane with side chains colored by atom: carbon (gray), oxygen (red) and nitrogen (blue). The TRP helix has been removed for clarity. (**C**) Close-up views of the $PIP_2$ binding site, with side chain coloring as in B. For clarity, helices without binding pocket residues have been hidden in panels B and C.

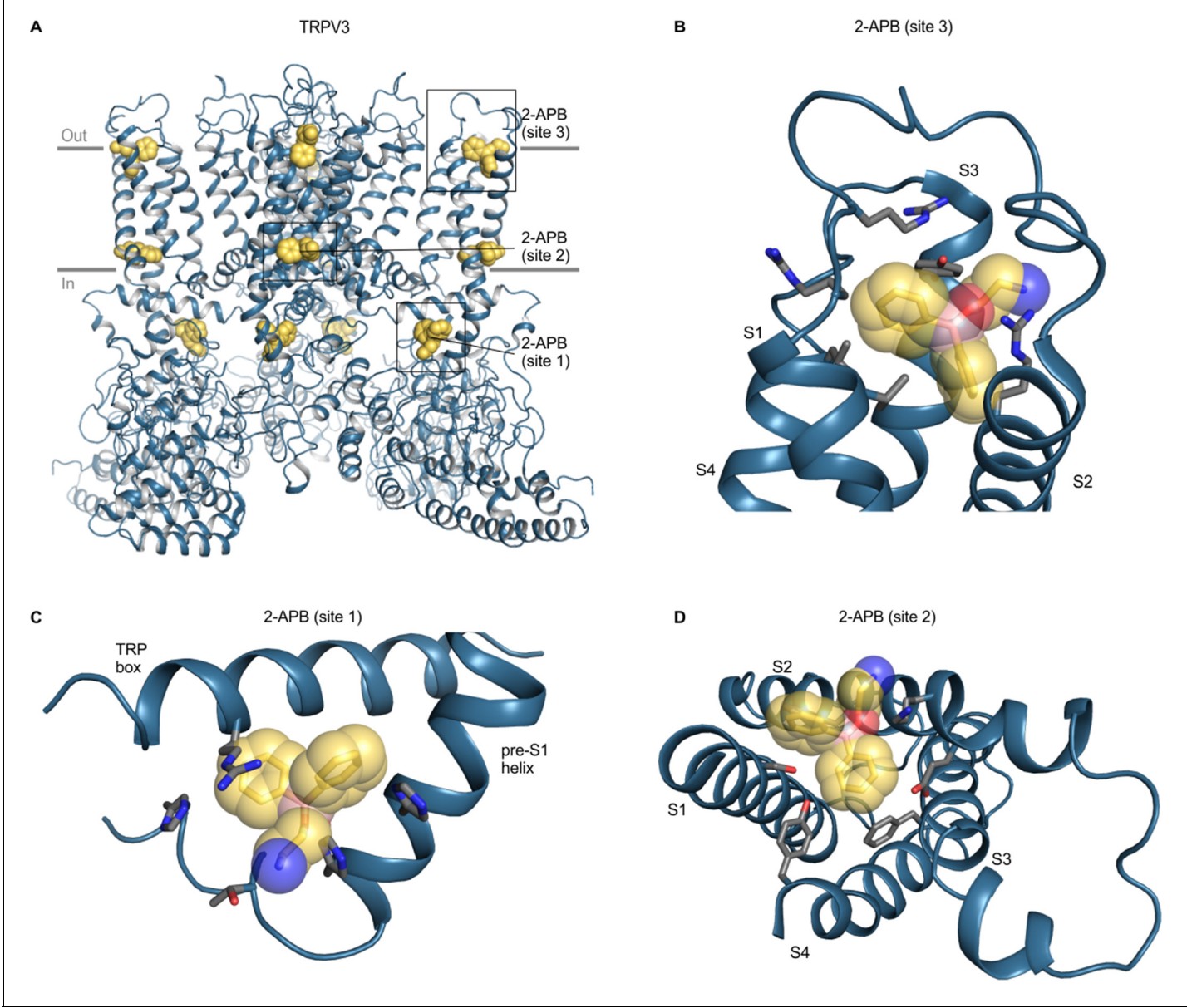

**Figure 9.** Structure of TRPV3 with 2-APB bound. (**A**) Structure of 2-APB bound TRPV3 (6dvz), with ligands shown as yellow spheres. (**B–D**) Close-up views of the three 2-APB-binding sites with side chains colored by atom: carbon (gray), oxygen (red) and nitrogen (blue). Boron atoms in 2-APB are colored in pink. Close-up in D is shown from the intracellular side of the membrane from same point of view as *Figure 8B*. For clarity, helices without binding pocket residues have been hidden in panels B-D.

The online version of this article includes the following figure supplement(s) for figure 9:

**Figure supplement 1.** Ligand-binding sites in TRPV2 and TRPML1.

confirmed using a brominated derivative of 2-APB (*Singh et al., 2018c*). Mutation of a conserved Tyr to Ala in site two increases the apparent affinity of 2-APB to TRPV1-3 and to TRPV6, even though the ligand is an activator in TRPV1-3 and an inhibitor in TRPV6 (*Singh et al., 2018c*). In other TRP channels, ligands have been observed to bind to pockets similar to site 2 for 2-APB binding in TRPV3, including TRPM8 antagonist AMTB (6o6r), TRPM8 antagonist TC-1 2014 (6o72), TRPC6 antagonist AM-1473 (6uza), and TRPC5 inhibitor clemizole (*Bai et al., 2020*; *Diver et al., 2019*; *Song et al., 2020*). Site three in mouse TRPV3 is located toward the extracellular side of the protein between S1 and S3 helices, and point mutations in this site decrease 2-APB affinity (*Singh et al., 2018a*; *Figure 9B*). Intriguingly, only point mutations in site one appear to have effects on 2-APB

specifically when compared to camphor (*Zubcevic et al., 2019a*). Using our structural alignments, we examined the conservation of all three sites, and found that all three are poorly conserved in other TRP channels (including TRPV6) when compared to TRPV3, and that the 2-APB site in TRPV6 only shows conservation with TRPV5 (*Figure 6*; *Figure 6—figure supplement 2*). This lack of conservation is surprising given that 2-APB can modulate the activity of many different TRP channels (as an agonist for TRPV1-3 and TRPM6; as an inhibitor of TRPM2, TRPM3, TRPM8, TRPC5, TRPC6 and TRPV6; as an inhibitor at low concentration and as agonist at high concentration for TRPM7) (*Chokshi et al., 2012*; *Colton and Zhu, 2007*; *Hu et al., 2004*; *Kovacs et al., 2012*; *Togashi et al., 2008*; *Xu et al., 2005*). Indeed, although specific side chains are poorly conserved, all three 2-APB sites contain multiple Arg, His and hydrophobic residues (*Figure 9*), suggesting that the structural basis for 2-APB binding (and activity) may rely more on side chain character than on binding pocket shape. Notably, 2-APB has been reported to undergo chemical changes in solution and adopt different pH-dependent configurations, such that different forms might bind to distinct sites or modulate channels differently (*Gao et al., 2016a*). It would be interesting to further explore potential 2-APB-binding sites in other TRP channels with both structural and mutagenesis approaches to better understand the promiscuous and pleiotropic behavior of this ligand.

The final ligand-binding site that we considered in the TM domain is located adjacent to the vanilloid binding pocket, at the interface between the S6 helix from one subunit and the neighboring pore loop, S5, and S6 helices. This pocket is lined by hydrophobic residues, including bulky side chains like Phe and Tyr. The phyto-cannabinoid cannabidiol (CBD) was identified in this pocket in structures of rat TRPV2 (*Pumroy et al., 2019*), and the agonist ML-SA1 was found in the corresponding location in human TRPML1 (*Fine et al., 2018*; *Schmiege et al., 2017*; *Figure 9—figure supplement 1*). For both channels, the activity of the ligand is modified by mutations in the identified binding pocket (*Fine et al., 2018*; *Pumroy et al., 2019*; *Schmiege et al., 2017*). Densities at similar sites have been observed for other TRP channel ligands, including TRPA1 inhibitor A-967079 (EMD-6268), TRPV5 inhibitor ZINC17988990 (6pbe), TRPV5 inhibitor ZINC9155420 (6pbf), TRPC5 inhibitor HC-070, and TRPC6 agonist AM-0883 (6uz8), and several of these interactions have been investigated further with mutagenesis (*Bai et al., 2020*; *Brewster and Gaudet, 2015*; *Hughes et al., 2019*; *Paulsen et al., 2015*; *Song et al., 2020*; *Ton et al., 2017*; *Woll et al., 2017*). The hydrophobic character of this pocket is relatively well conserved across TRP channels, with the pocket in the TRPML1 channels containing more polar residues, a feature that is conserved in other TRPML subfamily members that are also sensitive to ML-SA1 (*Figure 6*; *Figure 6—figure supplement 2*). Although the activity of CBD has not been widely explored across TRP channels, the ligand modulates the activity of many different ion channel proteins (*Ghovanloo et al., 2018*; *Hassan et al., 2014*; *Mahgoub et al., 2013*; *Qin et al., 2008*; *Ross et al., 2008*; *Thompson and Kearney, 2016*), consistent with binding to a hydrophobic cavity that opens to the lipid bilayer.

## Unique secondary structural elements within TM helices in TRP channels

The S1-S6 TM segments in all TRP channels adopt α-helical secondary structure (3.6 residues per turn) over most of their length. Alternative helical conformations in proteins include the $3_{10}$ helix that comprises three residues per turn and is thus more tightly wound (*Riek et al., 2001*), and the π helix that contains 4.6 residues per turn, creating a less-tightly wound bulge (*Riek and Graham, 2011*; *Riek et al., 2001*). Although the role of alternate helical conformations is not well understood, the presence of $3_{10}$ helices within the S4 helices of voltage-activated ion channels is thought to play an important role in the process of voltage sensing because $3_{10}$ helices position basic residues in different environments compared to an α helix (*Long et al., 2007*). In TRP channels, bulging helical sections resembling π helices have been commonly observed at the intracellular end of S6 helices and proposed to serve as hinges that facilitate opening of the S6 gate (*Kasimova et al., 2018*; *McGoldrick et al., 2018*; *Palovcak et al., 2015*; *Zubcevic et al., 2016*; *Zubcevic and Lee, 2019c*). Using our structural alignment and the DSSP algorithm (*Kabsch and Sander, 1983*; *Touw et al., 2015*), we assigned secondary structure to each residue in helices S1-S6 for all TRP channels in our alignment (*Figure 10*). Although the resolution of most available structures precludes a fine-grained analysis of individual residues, we nevertheless observed a couple of notable trends. A high frequency of π-helix-like elements within a relatively narrow region of S6 is readily apparent, consistent with conclusions from a recent review on the role of this alternative helix conformation in opening of TRP channels (*Zubcevic and Lee, 2019c*). However, it was not clear whether the presence of these

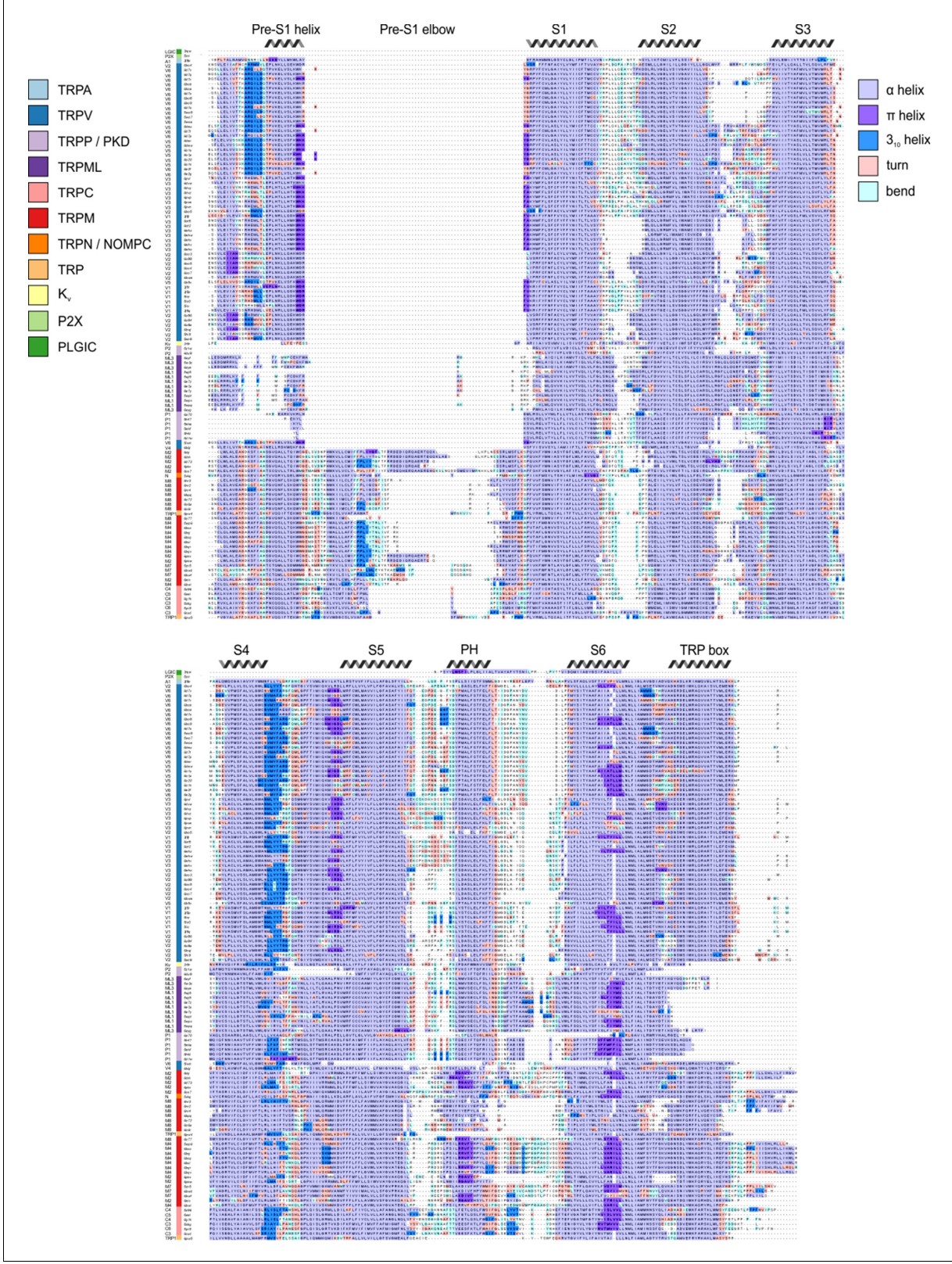

**Figure 10.** Alternate helical conformations with the TM segments of TRP channels. Structure-based sequence alignment, with residues colored based on secondary structure assigned by the DSSP algorithm. Sequences are ordered based on hierarchical clustering from *Figure 2*. Segments, including TM helices and other regions identified in *Figure 1F*, are labeled based on α-helicity consensus. PH stands for pore helix.

*Figure 10 continued on next page*

*Figure 10 continued*

The online version of this article includes the following source data and figure supplement(s) for figure 10:

**Source data 1.** Data file for S6 radius and consecutive S6 π helices scatterplot.

**Figure supplement 1.** Relationships between alternate helix conformations and the radius of the internal pore in TRP channels.

π-helical residues in the S6 helix correlated with changes in pore size, as might be expected for such a gating mechanism. Taking all structures together, there is a slight positive correlation between pore radius and the length of the S6 π helices (*Figure 10—figure supplement 1*). Analysis of correlations between pore radius and length of S6 π helices must be interpreted cautiously due to the lack of datapoints with large S6 pore radii, and the lack of short π helices, which arises from the DSSP definitional requirement of at least two helical turns. From the present analysis, we can say that structures with small pore radii are observed to have either fully α-helical or partially π-helical character in the S6 region. Further structural studies in search of states with wider pores will be needed to clarify the role of π helices in gating.

We also found that $3_{10}$ helices are commonly found at the intracellular end of the S4 helices across all TRP channel subfamilies (*Figure 10*), similar to what has been observed with the corresponding TM helices in voltage-activated ion channels. Notably, these $3_{10}$ helices within S4 are observed regardless of whether structures were determined in the presence or absence of activators (*Figure 10*). Finally, our analysis detects π helices within the S4-S5 linker and S5 helices of TRP channels spatially close to where they are observed in S6 helices, in particular in the TRPV subfamily of TRP channels (*Pumroy et al., 2019*; *Zubcevic et al., 2018a*; *Zubcevic et al., 2019b*; *Zubcevic et al., 2018b*). The presence of alternative helical conformations within TM regions of TRP channel structures is interesting and should motivate further exploration of their functional roles. Although limitations in resolution and structural model quality preclude using this analysis to support conservation of specific types of secondary structure elements between structures, it suggests conserved regions where structures are likely deviating from an α helix, possibly due to conformational flexibility.

## Discussion

The goal of the present study was to construct a structure-based alignment of the TM domains of available TRP channel structures using a uniform approach that allows systematic comparison of structural features in functionally important regions. Our analysis strongly supports the prevailing view that the intracellular end of the S6 helices forms a constriction or gate that prevents ion permeation in closed or non-conducting desensitized states, with the narrowest constrictions occurring at one of four positions along the S6 helices. It remains to be determined if the formation of some of these constrictions is specific to certain TRP channel subtypes or functional states (apo closed vs. desensitized or closed-sensitized). In addition, global analysis of the dimensions of the S6 gate region lead us to suggest that the open states for most TRP channels remain to be elucidated. The internal pore of zebrafish TRPM2 when bound by $Ca^{2+}$ and ADP ribose (6drj) has a radius of 4.4 Å and thus is likely large enough to permit rapid diffusion of hydrated cations, consistent with this structure representing an open state. The internal S6 region for rat TRPV1, mouse and human TRPV3 and rabbit TRPV5 are between 3 and 3.3 Å, which does not seem quite open enough to support permeation of hydrated cations (with large single channel conductance) or the entry of large quaternary ammonium ion blockers. A similar conundrum has been raised by the structures of many $K^+$ channels. That is, although the S6 gate regions of some $K^+$ channels have large radii consistent with an open state (e.g. 4.2 Å for Kv1.2/2.1 paddle chimera, 5 Å for hERG, 10 Å for Slo2 and 15 Å for hSlo1 with $Ca^{2+}$) (*Hite and MacKinnon, 2017*; *Long et al., 2007*; *Tao et al., 2017*; *Tao and MacKinnon, 2019b*; *Wang and MacKinnon, 2017*), in other cases the internal pores are narrower than expected (2.5 Å for Kir2.2, 3.5 Å for SK, 3–3.5 Å for GIRK2, 3 Å for KvAP and 2.5 Å for KCNQ1) (*Hansen et al., 2011*; *Lee and MacKinnon, 2018*; *Sun and MacKinnon, 2020*; *Tao and MacKinnon, 2019a*; *Whorton and MacKinnon, 2013*) under conditions expected to favor open states (*Figure 3—figure supplement 1*). It will be important to see whether structures of most TRP channels and some $K^+$ channels can be determined with more open S6 gates. Solving structures of open states of TRP

channels is particularly critical for understanding the structural basis by which different stimuli lead to channel opening, and we propose that future structural studies should focus on increasing construct open probability to facilitate a larger number of open-state particles on cryo-EM grids. However, the paucity of open structures among the currently available structures, many of which have been determined under conditions that increase channel open probability, suggests the existence of additional complexities in stabilizing TRP channel open states for structure determination. For example, the open probability of RTx-sensitive rat TRPV2 is nearly one as measured in functional experiments in cell membranes, but the structures of the rabbit orthologue of this construct solved in the presence of RTx do not appear sufficiently open to conduct ions (*Zhang et al., 2016*; *Zubcevic et al., 2019b*; *Zubcevic et al., 2018b*).

Another fascinating question concerns the mechanisms by which the external selectivity filters in TRP channels can select for monovalent cations (TRPM4 and TRPM5), divalent cations (TRPV5 and TRPV6) or support the permeation of both (all other TRP channels). The X-ray structure of rat TRPV6 (divalent selective) (*Saotome et al., 2016*) and a cryo-EM structure of mouse TRPM4 (monovalent selective) (*Guo et al., 2017* have led to interesting working hypotheses for these two classes of ion selectivity. In the case of TRPV6, divalent ions can be seen to bind within a narrow region of the filter that would require at least partial dehydration of the ion, suggesting that ion coordination and dehydration are critical to the mechanism of divalent ion selectivity. In the case of TRPM4, evidence of intersubunit hydrogen bonds within the filter lead to the proposal that the filter in monovalent selective TRP channels is structurally rigid and just large enough for hydrated monovalent ions to permeate. Although the filters of non-selective TRP channel have conserved features as noted earlier, the dimensions of the filters are remarkably varied when comparing structures within or between subfamilies and their lack of selectivity might suggest that both hydrated and dehydrated ions may permeate. Clearly higher resolution X-ray structures, where ion binding sites can be examined, will be needed to deduce the underlying mechanisms, and it will be critical to obtain evidence for whether ion permeation involves hydrated or dehydrated forms of permeant cations. Higher resolution structures will also facilitate molecular dynamics simulations to probe the energetics of ion permeation, including contributions from conformational flexibility and electrostatics of nearby charges.

The wealth of available TRP channel structures underscores the extent to which the S1-S4 domain, as well as the interface of this domain with the S5-S6 pore-forming domain and the TRP box functions as a hot spot for ligands to promote opening of TRP channels. This region includes the vanilloid-binding pocket in TRPV1, which is also hydrophobic in other TRP channels, perhaps reflecting a common lipid-binding site that regulates the activity of many TRP channels. The $Ca^{2+}$ and cooling-agent-binding sites in TRPM channels are also in close proximity and are relatively well conserved in the TRPM subfamily. Finally, sites 1 and 2 for the promiscuous regulator 2-APB are also positioned nearby, either below or above the TRP helix, respectively. Although the conservation of ligand-binding sites varies considerably across different TRP channels, it would be fascinating to attempt to engineer in ligand sensitivity into insensitive TRP channels to explore the extent to which gating mechanisms are related.

The binding of lipids to TRP channels and regulation of functional activity is a fascinating and emerging area in the field. Although we have not focused on lipid-binding sites because the quality of lipid-like densities is not high enough to identify the molecule definitively in most structures, there are a few notable exceptions. In cryo-EM structures of rat TRPV1 in nanodiscs, several well-defined phospholipid densities can be seen to interact simultaneously with the external membrane-exposed surface of the protein and the tarantula toxin DkTx (*Gao et al., 2016b*). In an apo structure of TRPV1 in nanodiscs (5irz), as well as structures of TRPC4 (5z96), TRPM2 (6co7), TRPM4 (6bwi, 6bqr, 6bqv), TRPM7 (5z × 5, 6bwd), NOMPC (5vkq), TRPP1 (5mke, 5mkf), TRPV5 (6dmr, 6dmu) and TRPV6 (6bo8) lipid density can be seen in the vanilloid-binding pocket (*Autzen et al., 2018*; *Duan et al., 2018a*; *Duan et al., 2018b*; *Duan et al., 2018c*; *Gao et al., 2016b*; *Hughes et al., 2018b*; *Jin et al., 2017*; *McGoldrick et al., 2018*; *Wilkes et al., 2017*; *Zhang et al., 2018*). Finally, a well-resolved molecule of PIP$_2$ can also be seen in flycatcher TRPM8 channels close to where $Ca^{2+}$ and cooling agents bind, and involving basic residues in the pre-S1 helix, the S4-S5 linker, the TRP domain and the cytoplasmic MHR4 domain (*Figure 8C*; *Yin et al., 2019a*). Residues lining this PIP$_2$ binding pocket are conserved in TRPM, TRPC, and, to a lesser extent, TRPV channels (*Figure 6*; *Figure 6—figure supplement 1*), although PIP$_2$ binding in TRPM8 involves some architectural elements, including the pre-S1 elbow domain and TRPM homology domains, that are unique to TRPM channels.

Lipid-like density was observed at similar sites in TRPM2 and algal TRP1 (*McGoldrick et al., 2019*; *Zhang et al., 2018*). $PIP_2$ is required for activation of both TRPM2 and TRPM8; indeed, exogenous $PIP_2$ analogs are sufficient to activate the TRPM8 channel at room temperature (*Liu and Qin, 2005*; *Yudin and Rohacs, 2012*).

We undertook a global alignment of TRP channel TM-domain structures to explore those features that are common to all TRP channels and those that may be unique to specific subfamily members. At the time we stopped adding structures to our alignment, there were 136 structures published over a 6-year period. Although this represents an unparalleled number of related ion channel structures to work with, we were surprised that our analysis identifies the need for additional structures, even for the TRPV and TRPM subtypes that dominate our structural alignment. In addition, for most TRP channels, it seems that fully open states have yet to be determined. We need additional structures of TRPM4 and TRPM5 to test mechanisms of monovalent cation selectivity, structures of TRPV5 and TRPV6 to interrogate mechanisms of divalent ion selectivity, and more structures bound to promiscuous modulators such as 2-APB. Those structures that have thus far been determined in lipid nanodiscs have begun to reveal key structural and functional roles of membrane lipids, and this is a particularly important area for further exploration.

## Materials and methods

### TRP channel structure selection

All TRP channel structures were identified by searching the PDB using the query 'TRP channel' on October 31st, 2019 (*Berman et al., 2000*). Structures with resolution poorer than 5 Å, as well as most non-domain-swapped mutant structures were excluded. Structures available from OPM (Orientation of Proteins in Membranes) were pulled from that database, and those that were not already available were analyzed using the PPM (Positioning of Proteins in Membranes) server (*Lomize et al., 2012*). The available cryo-EM structures only approach atomic resolution as determined by Fourier shell correlation, and EM electron density maps vary in quality in different regions. However, due to the large number of structures, comparing the structures collectively decreases the impact of random errors in model fitting due to insufficient density map resolvability.

### Structure file processing and domain definitions

Only transmembrane domains were used for alignment, so intra- and extra-cellular domains were identified and stripped. For structural alignments, several different regions of the proteins were defined, as follows. TM domains were defined as residues from the start of the pre-S1 domain to the end of the TRP box as determined by visual inspection of the structures (see *Figure 2—source data 1* for exact residues used). Pore domain definitions included all residues from the start of the S5 helix to the end of the S6 helix based on the results from OPM or PPM (see above). The S1-S4 domain was defined as all residues from the start of the S1 helix to the end of the S4 helix, based on OPM-identified TM segments. To exclude extramembranous domains, any loop connecting two OPM-defined TM segments with >100 residues was truncated to leave only the 10 residues on each side of the loop nearest to the TM segments. HETATOM entries were also removed. Alignments of the TM domain or pore domain included the entire tetrameric assembly. Prior to alignment, the ordering of the chain identifiers was standardized (counterclockwise as viewed from the extracellular side of the membrane), and chains were then combined into a single chain for compatibility with Fr-TM-Align. Alignments of S1-S4 domains included a single protomer, with all other chains deleted. Non-TRP channels were processed similarly (see *Figure 2—source data 1* for exact residues used). Structures were also categorized qualitatively into groups based on subfamily, experimental method, sample conditions, and ligand-binding state (see *Figure 2—source data 1* for category assignments).

### Structure-based alignment

To obtain a structure-based, sequence-agnostic sequence alignment, structures were first aligned pairwise using Fr-TM-Align version 1.0, a fragment-based alignment approach that aligns residues based on patterns of secondary structure (*Pandit and Skolnick, 2008*). Fr-TM-Align has been tested on membrane proteins and is robust even to large conformational changes (*Stamm and Forrest,*

*2015*). As with other methods, the aligned structures are iteratively aligned and scored for alignment match before the alignment with the best pairwise TM-score is chosen. The TM-score is a length-independent analogue of RMSD, and indicates global protein fold similarity, with 1.0 indicating identical structures and an average of 0.3 for randomly selected proteins, where TM-scores above 0.6 indicate a common fold (*Xu and Zhang, 2010*; *Zhang and Skolnick, 2004*). Fr-TM Align also reports the transformation matrix for each pairwise structural alignment. TM-scores are normalized to the length of the stationary protein in the pairwise mobile-stationary alignment, resulting in asymmetrical scores depending on which protein of the pair is used as the mobile structure and which as the stationary structure. Therefore, Fr-TM-Align was performed twice for each pair of proteins, exchanging the mobile and stationary structures. Mobile and stationary proteins are represented along the vertical and horizontal axes, respectively, in the heatmaps of *Figure 2* and *Figure 2—figure supplements 2* and *3*.

## Clustering

Clustering was performed along the stationary axis in the TM-score heatmap. TM-scores were converted into pseudo-distance scores where: TM-distance = 1 – TM-score, and hierarchical clustering based on TM-distance was calculated with Seaborn's clustermap function using the Nearest Point Algorithm in Euclidean space (parameters: method='single', metric='euclidean', Seaborn version 0.9.0) (*Müllner, 2011*; *Waskom et al., 2018*).

## Creating structure-based multiple sequence alignments

Residues considered in the TM domain alignment were used to build multiple sequence alignments. One sequence was chosen as the reference (nvTRPM2, 6co7), while all other proteins were added according to their pairwise alignment with the reference using pyali version 0.1.1 (*Tang, 2019*). Residues that did not align with an amino acid in the reference structure, that is insertions, were omitted from the multiple sequence alignment.

## Creating sequence-based multiple sequence alignment

The same amino acid sequences used for the structure-based alignment were aligned with ClustalO-mega using default settings (*Madeira et al., 2019*). To enable comparison between structure-based and sequence-based alignments, any residues that did not align with an amino acid in 6co7 were omitted from the sequence-based alignment.

## Determining secondary structure and pore radius

Pore dimensions were estimated using HOLE version 2.0, which reports, for each point along the length of the pore, the radius of the largest sphere that can be fit in the pore without intersecting with a neighboring atom, as defined by its van der Waals radius (*Smart et al., 1996*). Hydrogen atoms were not considered in this analysis. Residues were identified as lining the pore if the distance between any of its atoms and the axis of the HOLE profile was equal to the sum of the van der Waals radius of that atom and the pore radius at that point. The minimum pore radius for a given residue is defined as the smallest radius of the HOLE plot assigned to any atom in that residue.

The DSSP algorithm version 3.0.0 was used to assign the secondary structure of each residue of the protein (*Kabsch and Sander, 1983*; *Touw et al., 2015*).

## Identifying and analyzing selectivity filters

Selectivity filters were determined by visual inspection and consensus among structures (*Figure 4*). Selectivity filters were compared pairwise for all structures, with percent identity determined by the number of identical residues, excluding gaps, in equivalent positions. Similarity was defined by a positive score in the BLOSUM62 matrix (*Henikoff and Henikoff, 1992*).

$$Identity_\% = 100\% * \frac{n_{identical}}{n_{ref}}$$

$$Similarity_\% = 100\% * \frac{n_{similar}}{n_{ref}}$$

## Identifying and analyzing ligand binding pockets

Structures lacking ligands were considered to be in their apo states. For structures that contained ligands, any amino acid with any side-chain atom within 4 Å of the ligand molecule was considered part of the ligand-binding pocket. Equivalent residues in other structures were identified using the structure-based multiple sequence alignment. To calculate percent identity and similarity of the binding pocket residues, one ligand-bound structure was chosen to provide the reference ligand-binding pocket motif, and binding pockets from all other structures were analyzed to determine the percentage of residues that were identical or similar to those in the equivalent position in the reference. Identity and similarity were defined as for selectivity filters, above.

## Generating figures

All figures of protein structures were created after aligning each structure to the reference structure (nvTRM2, 6co7) using Fr-TM-Align as described above. For visualization of the entire structure, the corresponding transformation matrix was reapplied in PyMOL version 2.2.3 (*Schrödinger, 2015*). Analysis and visualization were performed in Python 3.6.7 using Anaconda 5.2.0 packages: SciPy 1.1.0, Matplotlib 2.2.2, pandas 0.24.2, seaborn 0.9.0, Numpy 1.14.3, pyali 0.1.1, HOLE 2.0 implemented with MDAnalysis 0.18.0, DSSP 3.0.0 and Biopython 1.72 (*Cock et al., 2009*; *Hamelryck and Manderick, 2003*; *Hunter, 2007*; *McKinney, 2010*; *Schrödinger, 2015*; *Virtanen et al., 2020*; *Waskom et al., 2018*). All sequence alignments were visualized with Jalview 2.10.5 (*Waterhouse et al., 2009*).

## Code and dataset availability

The code necessary to replicate this data and analysis is available on GitHub (*Huffer, 2020* https://github.com/kehuffer/TRP_Structural_Alignment copy archived at https://github.com/elifesciences-publications/TRP_Structural_Alignment). Much of the data from the analysis, including Fr-TM-Align outputs, PDB files of each TRP channel structure aligned to the reference structure (nvTRPM2, 6co7), and multiple sequence alignments using each structure as the reference structure, are available from Zenodo (https://zenodo.org/record/3972100#.XzPkNPlKguU).

# Acknowledgements

We thank Joe Mindell, Mark Mayer, José Faraldo Gómez, Karen Fleming and members of the Swartz laboratory for helpful discussions.

# Additional information

### Competing interests

Kenton J Swartz: Senior editor, *eLife*. Lucy R Forrest: Reviewing editor, *eLife*. The other authors declare that no competing interests exist.

### Funding

| Funder | Grant reference number | Author |
| --- | --- | --- |
| National Institute of Neurological Disorders and Stroke | NS002945 | Kenton J Swartz |
| National Institute of Neurological Disorders and Stroke | NS 003139 | Lucy R Forrest |
| National Institute of Neurological Disorders and Stroke | K99 Pathway to Independence Award | Andrés Jara-Oseguera |

The funders had no role in study design, data collection and interpretation, or the decision to submit the work for publication.

## Author contributions
Katherine E Huffer, Conceptualization, Data curation, Formal analysis, Validation, Investigation, Visualization, Methodology, Writing - original draft, Writing - review and editing; Antoniya A Aleksandrova, Conceptualization, Formal analysis, Supervision, Validation, Investigation, Methodology, Project administration, Writing - review and editing; Andrés Jara-Oseguera, Conceptualization, Validation, Investigation, Writing - review and editing; Lucy R Forrest, Conceptualization, Resources, Supervision, Funding acquisition, Validation, Investigation, Methodology, Project administration, Writing - review and editing; Kenton J Swartz, Conceptualization, Resources, Supervision, Funding acquisition, Validation, Investigation, Methodology, Writing - original draft, Project administration, Writing - review and editing

## Author ORCIDs
Katherine E Huffer ![ORCID] https://orcid.org/0000-0001-5003-3140
Antoniya A Aleksandrova ![ORCID] http://orcid.org/0000-0001-7393-1787
Andrés Jara-Oseguera ![ORCID] http://orcid.org/0000-0001-5921-9320
Lucy R Forrest ![ORCID] http://orcid.org/0000-0003-1855-7985
Kenton J Swartz ![ORCID] https://orcid.org/0000-0003-3419-0765

## Decision letter and Author response
Decision letter https://doi.org/10.7554/eLife.58660.sa1
Author response https://doi.org/10.7554/eLife.58660.sa2

# Additional files

## Supplementary files
• Transparent reporting form

## Data availability
All data generated or analyzed during this study are included in the manuscript and supporting files. Source data files have been provided for Figures 2, 3, 4, 6, and 10. The code necessary to reproduce this data and analysis is available on GitHub (https://github.com/kehuffer/TRP_Structural_Alignment copy archived at https://github.com/elifesciences-publications/TRP_Structural_Alignment).

The following previously published dataset was used:

| Author(s) | Year | Dataset title | Dataset URL | Database and Identifier |
|---|---|---|---|---|
| Katherine E Huffer, Antoniya A Aleksandrova, Andrés Jara-Oseguera, Lucy R Forrest, Kenton J Swartz | 2020 | Underlying data for "Global alignment and assessment of TRP channel transmembrane domain structures to explore functional mechanisms" | https://zenodo.org/record/3972100#.XzKb5kF7kuU | Zenodo, 10.5281/zenodo.3972100 |

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
