## [Decision Letter]

**Acceptance summary:**

This study is an impressive compilation of structural information available on the transmembrane domain architecture of the entire TRP channel family, based on structural alignments of 120 atomic resolution TRP channel structures. The comparison focuses on the channel gate, the selectivity filter, the ligand binding sites, and a pi-helical bulge in the S6 helix. The analysis identifies structural motifs conserved across all seven TRP channel subfamilies, as well as motifs that are subfamily specific. These comparisons help to understand the mechanistic basis of mono- vs. divalent selectivity or non-selectivity and reveal conserved structural features of ligand binding sites. The global viewpoint of this manuscript makes it a unique contribution to the TRP field, appealing to a broad readership.

**Decision letter after peer review:**

Thank you for submitting your article "Global alignment and assessment of TRP channel transmembrane domain structures to explore functional mechanisms" for consideration by *eLife*. Your article has been reviewed by three peer reviewers, including László Csanády as the Reviewing Editor and Reviewer #1, and the evaluation has been overseen by Richard Aldrich as the Senior Editor. The following individuals involved in review of your submission have agreed to reveal their identity: Juan Du (Reviewer #2); Lejla Zubcevic (Reviewer #3).

The reviewers have discussed the reviews with one another and the Reviewing Editor has drafted this decision to help you prepare a revised submission.

Summary:

This study is an impressive compilation of structural information available on the transmembrane domain architecture of the entire TRP channel family, based on structural alignments of 120 atomic resolution TRP channel structures. The comparison focuses on the channel gate, the selectivity filter, the ligand binding sites, and a pi-helical bulge in the S6 helix. The analysis identifies structural motifs conserved across all seven TRP channel subfamilies, as well as motifs that are subfamily specific. These comparisons help to understand the mechanistic basis of mono- vs. divalent selectivity or non-selectivity in various TRP channels. They also reveal that structural features of binding sites for ligands that specifically act on one subfamily are nevertheless conserved in other subfamilies. This suggests that – despite a diverse range of activating stimuli – the basic mechanism of gating is conserved among all TRP channels, and raises the possibility of further layers of channel regulation by as yet unidentified modulators.

Finally, the study clearly shows that more structures are needed to fully understand the movements that lead to gate opening in TRP channels. Whereas most published structural studies have focused on individual TRP subfamilies, the global viewpoint of this manuscript makes it a unique contribution to the TRP field, appealing to a broad readership.

The reviewers have noted no major concerns, but have compiled a number of smaller issues, which should be easily addressed by the authors to enhance the clarity and comprehensiveness of the presentation.

Essential revisions:

1) The authors should specify the organism of the TRP channels and the PDB code of the structures in the text. It would be nice to add protein name and PDB code directly in the figures rather than in the figure legends.

2) "the S6 gate region of TRPM2 (4.4A radius) (Zhang et al., 2018a)": The citation for the ADPR- and Ca-bound zebrafish TRPM2 in open state is incorrect – it should be Huang et al., 2018. The TRPM2 structure from *Nematostella vectensis* in Zhang et al., 2018a, as cited by the authors here, is in a calcium-bound closed state. It is also important to add the organisms as the human TRPM2 in the same ADPR + Ca bound ligand condition by Huang et al., 2019, shows a closed pore.

3) The authors have discussed determinants of ion selectivity among TRP channels by looking into their selectively filters. For channels in the TRPM subfamily, a Gln residue is crucial for the selectivity against monovalent cations (in TRPM4 and TRPM5). However, it should also be noted in the manuscript that the geometry of the selectivity filter may also play a role. For instance, despite having a Gln, TRPM2 is non-selective likely owing to its flat selectivity filter (Huang et al. 2020 Cell Calcium, Huang et al., 2018). A few references to the structures of TRPM4 (Winkler et al., 2017, Autzen et al., 2017, Duan et al., 2018) are missing in the selectivity section and in the Discussion. The authors may also provide insights on why TRPM8 is non-selective despite having a Gln based on the structure by Diver et al., 2019.

4) The new TRPA1 structures by Zhao et al. 2019 bioRxiv with an open structure should be added into panel A in Figure 3—figure supplement 2.

5) We suggest changing the title of the section "Ligand-binding pockets in TRP channels" to "Ligand-binding pockets in the TMDs of TRP channels" because the focus here is limited to the ligands in the TMDs. The authors should cite the structure of TRPA1 (Zhao et al. 2019 bioRxiv), which also contains a Ca binding site in the TMD; this Ca plays a role in channel desensitization. Furthermore, several drug and lipid binding sites have been defined in the TMD of TRPC channels, such as Bai et al., 2020, Tang et al., 2018, Song et al., 2020, which should be cited and mentioned. The citations for the first sentence in the fifth paragraph should be Autzen et al., 2018, Diver et al., 2019, Zhao et al., 2019, Huang et al., 2018, Yin et al., 2019a, 2018, Zhang et al., 2018a, Huang et al., 2019.

6) The authors have selected the models used in the study based on resolution. However, there is also a huge variability in model quality amongst the published structures (i.e. how the models fit into the cryo-EM maps, quality of the geometry of the models, etc.) that could impact the data analysis, especially the parts concerning ligand binding and secondary structure. The authors should comment on this in the manuscript.

7) "For all TRP channel structures, the open probability of the construct used for structure determination has not been measured in either the absence or presence of activating stimuli, hindering objective attempts to relate specific structures to distinct functional states." "Solving structures of open states of TRP channels is particularly critical for understanding the structural basis by which different stimuli lead to channel opening, and we propose that future structural studies should focus on increasing construct open probability to facilitate a larger number of open-state particles on cryo-EM grids."

We are not convinced that these two measures would necessarily correlate well in TRP channels. For example, the Po of the RTx-sensitive TRPV2 was measured by the Swartz lab and found to be ~1 and yet the structure of the construct in the presence of RTx was not captured in the fully open state. Also, the open state structure of the constitutively open TRPV6 could only be achieved by introducing a mutation. The assumption that there would be a direct correlation between the Po and the probability of capturing the open state of these polymodal channels might be somewhat simplistic. There could be factors at play in stabilizing the open structure of TRPs that we are yet to establish. The absence of bona fide fully open structures amongst the >100 PDB deposits points in that direction.

8) "In addition, in the Ca^2+^-icilin complex the intracellular end of the S4 helix adopts an alternate conformation that repositions residues in the binding pocket, a difference that is not seen for WS-12, suggesting that different cooling agents have distinct mechanisms of activation."

It would perhaps be more accurate to say that they "might have distinct mechanisms of activation". The Yin et al., 2019, study shows that PIP_2_ in the WS-12-bound structure is not fully engaged which could be an alternative explanation for why the S4 helix remains α-helical. In addition, PIP_2_ is necessary for TRPM8 function and since the PIP_2_ site is only complete when the S4 helix is 3-10 helical, this transition in S4 may be a prerequisite for channel activation by both WS-12 and icilin.

9) "A correlate of this proposal is that the apo form would not conduct ions; although the selectivity filters in apo structures of TRPV1 and TRPV2 would be too narrow for hydrated ions to permeate, the dimensions would likely be sufficient for partially dehydrated ions to move through the filter…. Lending support to ion permeation through narrow selectivity filters are K^+^ channels, where ion dehydration is thought to be central to the mechanism of ion selectivity (Doyle et al., 1998; Zhou et al., 2001) and for which the minimal radii within the selectivity filter are <1.0 Å for structurally-conserved selectivity filters in different channels."

It seems important to note that the chemistry of the selectivity filters of TRPV1 and TRPV2 and those of K^+^-selective channels are very different. The narrow filter of the K^+^ channel is critical for the energetics of conducting dehydrated K^+^ ions. In apo TRPV1 and TRPV2, the selectivity filter is not only narrow but hydrophobically sealed by a methionine seal, making the apo conformation energetically unfavorable for ion passage.

While the study by Jara-Oseguera et al., 2019 suggested that the selectivity filter of TRPV1 and TRPV2 does not act as an activation gate, it did find that conformational changes do occur during activation and that the cytoplasmic gate is allosterically coupled to the selectivity filter. This seems to be an important part of the story that should be included.

10) "Site 1 in TRPV3 is located within the cytoplasm at the interface between the TRP helix and the pre-S1 helix (Figure 9C) and mutations in this site also alter the apparent affinity for 2-APB (Singh et al., 2018a; Zubcevic et al., 2019a). "

It'd be appropriate to also cite Hu et al., 2009.

11) Subsection “Ligand-binding pockets in TRP channels”, sixth paragraph: 2-APB residues in sites 1, 2 and 3 were tested electrophysiologically (Zubcevic et al., 2019). Only mutations in site 1 affected the 2-APB response.

12) "This PIP_2_ binding pocket is conserved in TRPM, TRPC, and, to a lesser extent, TRPV channels."

It might be more appropriate to state that some of the residues involved in PIP_2_ binding in TRPM8 are conserved in other TRPs. However, the quaternary structure of the pocket is not conserved: TRPV and TRPC channels do not have the same architectural elements as TRPM (the pre-S1 domain, the MHR4). And when comparing with other TRPM channels, the MHR4 in TRPM8 is positioned very differently which enables it to be a part of the PIP_2_ binding site. In addition, MHR4 residue K605 (not conserved in other TRPMs) has been shown to be critical for PIP_2_ binding (Yin et al., 2019).

13) Subsection “Unique secondary structural elements within TM helices in TRP channels”: What is the impact of resolution and model quality on the secondary structure analysis? Does the algorithm only recognize geometrically soundly built pi-helices? Can it mischaracterize some poorly built alpha-helices as pi-helices? What quality controls are implemented?

14) Check reference "Lipid-like density was observed at a similar site in TRPM2 (Yin et al., 2019a)". Did the authors mean Zhang et al., 2018?

15) Figure 2: The TM score used for generating this matrix is asymmetrical. Wouldn't it be more natural to define the TM score in a symmetrical fashion?

16) "The Ca^2+^ binding sites… involve… the S4-5 linker":

Should be S2-3 linker.

17) "pleotropic" should be pleiotropic.

18) "The internal pore of TRPM2… has a radius of 4.4A": please specify that you are referring to the zebrafish TRPM2 structure.

19) "PIP_2_ is thought to be required…": maybe "known to be required" would be more appropriate, as this has been demonstrated both for TRPM2 and for TRPM8.

20) "TRP channels structures": should be TRP channel structures.

21) Figure 1F: "extracytosolic" – maybe extracellular?

22) Figure 3A: Maybe align panel A to B and C in the vertical direction, to match the levels of sites A, B, C, and D. Maybe extend the horizontal lines also to panel A.

Figure 3—figure supplement 2C: The open TRPM2 pore profile (yellow line) is from zebrafish (6drj), whereas the closed profile (blue line) is from a low-resolution human TRPM2 structure (6mix). Maybe replace blue profile with the profile for the apo zebrafish structure (6drk)?

23) "white to blue": should be white to orange.

---

## [Author Response]

Essential revisions:1) The authors should specify the organism of the TRP channels and the PDB code of the structures in the text. It would be nice to add protein name and PDB code directly in the figures rather than in the figure legends.

We have added PDB IDs and organism names in multiple places in the text, as well as in Figure 1, Figure 3, and Figure 5.

2) "the S6 gate region of TRPM2 (4.4A radius) (Zhang et al., 2018a)": The citation for the ADPR- and Ca-bound zebrafish TRPM2 in open state is incorrect – it should be Huang et al., 2018. The TRPM2 structure from Nematostella vectensis in Zhang et al., 2018a, as cited by the authors here, is in a calcium-bound closed state. It is also important to add the organisms as the human TRPM2 in the same ADPR + Ca bound ligand condition by Huang et al., 2019 shows a closed pore.

Thanks for catching this error. We have revised this section of the text to cite the appropriate paper and to include the PDB ID and organism name for the structures discussed.

3) The authors have discussed determinants of ion selectivity among TRP channels by looking into their selectively filters. For channels in the TRPM subfamily, a Gln residue is crucial for the selectivity against monovalent cations (in TRPM4 and TRPM5). However, it should also be noted in the manuscript that the geometry of the selectivity filter may also play a role. For instance, despite having a Gln, TRPM2 is non-selective likely owing to its flat selectivity filter (Huang et al. 2020 Cell Calcium, Huang et al., 2018). A few references to the structures of TRPM4 (Winkler et al., 2017, Autzen et al., 2017, Duan et al., 2018) are missing in the selectivity section and in the Discussion. The authors may also provide insights on why TRPM8 is non-selective despite having a Gln based on the structure by Diver et al., 2019.

We have revised this section to make the points of the reviewers more clearly and have added the requested citations.

4) The new TRPA1 structures by Zhao et al. 2019 bioRxiv with an open structure should be added into panel A in Figure 3—figure supplement 2.

As we state in the manuscript, it was necessary to choose a cutoff date for structure inclusion, even though we continue to see a steady pace of interesting new TRP channel structures published since that cutoff. Adding new structures to the analysis would require re-running the structural alignment, re-generating the figures, assessing the new data, and updating the text. By end of that process, more structures would have been solved and the cycle would begin again. Recognizing that this is an active and exciting field, we have published the code necessary for analysis and multiple sequence alignment construction on GitHub in order to facilitate reanalysis including future new structures. In addition, we have released the key raw outputs of our work on Zenodo so that they can be easily examined and extended in the future. Having said that, we have cited and discussed any interesting new structures like TRPA1 that were not included in our analysis so the reader can otherwise be up to date.

5) We suggest changing the title of the section "Ligand-binding pockets in TRP channels" to "Ligand-binding pockets in the TMDs of TRP channels" because the focus here is limited to the ligands in the TMDs. The authors should cite the structure of TRPA1 (Zhao et al. 2019 bioRxiv), which also contains a Ca binding site in the TMD; this Ca plays a role in channel desensitization. Furthermore, several drug and lipid binding sites have been defined in the TMD of TRPC channels, such as Bai et al., 2020, Tang et al., 2018, Song et al., 2020, which should be cited and mentioned. The citations for the first sentence in the fifth paragraph should be Autzen et al., 2018, Diver et al., 2019, Zhao et al., 2019, Huang et al., 2018, Yin et al., 2019a, 2018, Zhang et al., 2018a, Huang et al., 2019.

We have changed the title of the section as suggested and have updated the text to include discussions of papers written since our structural inclusion cut-off date (even though we have not expanded our structural alignment analysis to include these figures as explained in point 4). Structures discussed are referenced by PDB ID, except where the structures are not yet available from the PDB.

6) The authors have selected the models used in the study based on resolution. However, there is also a huge variability in model quality amongst the published structures (i.e. how the models fit into the cryo-EM maps, quality of the geometry of the models, etc.) that could impact the data analysis, especially the parts concerning ligand binding and secondary structure. The authors should comment on this in the manuscript.

This is an important point, and we have added a few sentences at the beginning of the Results section to bring this issue to the attention of the reader.

7) "For all TRP channel structures, the open probability of the construct used for structure determination has not been measured in either the absence or presence of activating stimuli, hindering objective attempts to relate specific structures to distinct functional states." "Solving structures of open states of TRP channels is particularly critical for understanding the structural basis by which different stimuli lead to channel opening, and we propose that future structural studies should focus on increasing construct open probability to facilitate a larger number of open-state particles on cryo-EM grids."We are not convinced that these two measures would necessarily correlate well in TRP channels. For example, the Po of the RTx-sensitive TRPV2 was measured by the Swartz lab and found to be ~1 and yet the structure of the construct in the presence of RTx was not captured in the fully open state. Also, the open state structure of the constitutively open TRPV6 could only be achieved by introducing a mutation. The assumption that there would be a direct correlation between the Po and the probability of capturing the open state of these polymodal channels might be somewhat simplistic. There could be factors at play in stabilizing the open structure of TRPs that we are yet to establish. The absence of bona fide fully open structures amongst the >100 PDB deposits points in that direction.

We really appreciate this point and have revised the section in the Discussion to present a more nuanced appreciation of the challenges of solving open state structures while highlighting the need for more open state structures.

8) "In addition, in the Ca^2+^-icilin complex the intracellular end of the S4 helix adopts an alternate conformation that repositions residues in the binding pocket, a difference that is not seen for WS-12, suggesting that different cooling agents have distinct mechanisms of activation."It would perhaps be more accurate to say that they "might have distinct mechanisms of activation". The Yin et al., 2019, study shows that PIP_2_ in the WS-12-bound structure is not fully engaged which could be an alternative explanation for why the S4 helix remains α-helical. In addition, PIP_2_ is necessary for TRPM8 function and since the PIP_2_ site is only complete when the S4 helix is 3-10 helical, this transition in S4 may be a prerequisite for channel activation by both WS-12 and icilin.

We have added to the text to address this excellent point.

9) "A correlate of this proposal is that the apo form would not conduct ions; although the selectivity filters in apo structures of TRPV1 and TRPV2 would be too narrow for hydrated ions to permeate, the dimensions would likely be sufficient for partially dehydrated ions to move through the filter…. Lending support to ion permeation through narrow selectivity filters are K^+^ channels, where ion dehydration is thought to be central to the mechanism of ion selectivity (Doyle et al., 1998; Zhou et al., 2001) and for which the minimal radii within the selectivity filter are <1.0 Å for structurally-conserved selectivity filters in different channels."It seems important to note that the chemistry of the selectivity filters of TRPV1 and TRPV2 and those of K^+^-selective channels are very different. The narrow filter of the K^+^ channel is critical for the energetics of conducting dehydrated K^+^ ions. In apo TRPV1 and TRPV2, the selectivity filter is not only narrow but hydrophobically sealed by a methionine seal, making the apo conformation energetically unfavorable for ion passage.While the study by Jara-Oseguera et al., 2019 suggested that the selectivity filter of TRPV1 and TRPV2 does not act as an activation gate, it did find that conformational changes do occur during activation and that the cytoplasmic gate is allosterically coupled to the selectivity filter. This seems to be an important part of the story that should be included.

We really appreciate these comments and we have revised this section to provide a more nuanced description of what we currently understand and what open questions remain to be answered with additional structures and experiments.

10) "Site 1 in TRPV3 is located within the cytoplasm at the interface between the TRP helix and the pre-S1 helix (Figure 9C) and mutations in this site also alter the apparent affinity for 2-APB (Singh et al., 2018a; Zubcevic et al., 2019a)."It'd be appropriate to also cite Hu et al., 2009.

Updated, thank you.

11) Subsection “Ligand-binding pockets in TRP channels”, sixth paragraph: 2-APB residues in sites 1, 2 and 3 were tested electrophysiologically (Zubcevic et al., 2019). Only mutations in site 1 affected the 2-APB response.

This sentence has been revised to be accurate.

12) "This PIP_2_ binding pocket is conserved in TRPM, TRPC, and, to a lesser extent, TRPV channels."It might be more appropriate to state that some of the residues involved in PIP_2_ binding in TRPM8 are conserved in other TRPs. However, the quaternary structure of the pocket is not conserved: TRPV and TRPC channels do not have the same architectural elements as TRPM (the pre-S1 domain, the MHR4). And when comparing with other TRPM channels, the MHR4 in TRPM8 is positioned very differently which enables it to be a part of the PIP_2_ binding site. In addition, MHR4 residue K605 (not conserved in other TRPMs) has been shown to be critical for PIP_2_ binding (Yin et al., 2019).

These are excellent points and we have revised the text to provide the reader with a better understanding of the PIP_2_ binding pockets in TRP channels.

13) Subsection “Unique secondary structural elements within TM helices in TRP channels”: What is the impact of resolution and model quality on the secondary structure analysis? Does the algorithm only recognize geometrically soundly built pi-helices? Can it mischaracterize some poorly built alpha-helices as pi-helices? What quality controls are implemented?

The secondary structure elements (SSEs) were assigned with DSSP, the de facto standard for SSE assignment (Joosten et al., PMID 21071423), used also by the Protein Databank. DSSP assigns helices based on the strength of the electrostatic component of the hydrogen-bonding, determined using an empirical function depending on the distance and angle between backbone atoms (Kabsch and Sander, PMID 6667333). The assignment of α and pi-helices thus depends on whether the backbone atoms of residue *i* adopt a higher-energy hydrogen bond with the backbone atoms of residue *i+5*, than with the backbone atoms of the *i*+4 residue. Only when two or more of those types of H-bond are found sequentially will a segment be assigned either α or π (or 3-10) helix. The gradual nature of the H-bond assignment, rather than, say, using a cutoff distance, as well as the need for multiple assignments in a row, reduces the changes of mis-assignment due to subtle differences in atom positioning due to low resolution and/or poor model quality. In fact, historically, there has been a concern that algorithms such as DSSP under-assign pi-helices compared to α-helices (Cooley et al., PMID 20888342); this issue has been corrected in the recent release of DSSP used in this study (Touw et al., PMID 25352545).

Although we share the reviewers’ concern regarding the potential that some of the structures may be poorly built, the impact on the secondary structure assignments is hard to quantify because, to date, there is no widely-agreed upon measure in the field of how to assess resolution or model quality in local regions of the structure, especially in cryo-EM structures. Therefore, we find ourselves in a position to only assess the structures as they are, assuming that each author has done their best to build them, and after excluding the lowest-resolution structures. Since the pi-helices and 3-10 helices discussed in the manuscript are consistently found in the same regions of the structure, and since we discuss the observation of patterns rather than single-residue assignments, we expect that our overall conclusions will be unaffected by these modeling issues. Nevertheless, we trust that, by making the underlying code available, a more detailed analysis of the effect of local resolution on the specific assignments can be carried out relatively trivially once metrics of local cryo-EM model quality become standardized. We have revised the text to explicitly discuss the resolution cutoff and the likely impact of model quality (subsection “Structure-based alignment of TRP channels”; see also response to point #6), and to indicate that detailed analysis is precluded by the resolution of the structures (subsection “Unique secondary structural elements within TM helices in TRP channels”).

14) Check reference "Lipid-like density was observed at a similar site in TRPM2 (Yin et al., 2019a)". Did the authors mean Zhang et al., 2018?

Fixed, thank you.

15) Figure 2: The TM score used for generating this matrix is asymmetrical. Wouldn't it be more natural to define the TM score in a symmetrical fashion?

The TM-score is formally correct as asymmetrical based on the original definition (Zhang and Skolnick, 2004). Forcing the TM-score to be symmetric would require altering the scoring definition such that it would no longer be a TM-score, thus sacrificing the foundations of prior work on the meaning and utility of the TM-score (Xu and Zhang, 2010). Here, the TM-score asymmetry does not profoundly alter the clustering results depending on whether clustering is performed across stationary structures or across mobile structures, so we have used clustering along only the stationary structures for clarity. If a reader wishes to investigate a symmetrical metric, the RMSD values can be obtained for each pair of proteins using the code available on GitHub and Zenodo.

16) "The Ca^2+^ binding sites… involve… the S4-5 linker":Should be S2-3 linker.

Fixed, thank you.

17) "pleotropic" should be pleiotropic.

Fixed, thank you.

18) "The internal pore of TRPM2… has a radius of 4.4A": please specify that you are referring to the zebrafish TRPM2 structure.

This sentence has been clarified.

19) "PIP_2_ is thought to be required…": maybe "known to be required" would be more appropriate, as this has been demonstrated both for TRPM2 and for TRPM8.

This section has been re-worded as requested.

20) "TRP channels structures": should be TRP channel structures.

Fixed, thank you.

21) Figure 1F: "extracytosolic" – maybe extracellular?

We chose this word deliberately because some TRP channels are expressed at membranes other than the plasma membrane. In particular, TRPML channels are expressed in liposomes, thus making these domains luminal rather than extracellular.

22) Figure 3A: Maybe align panel A to B and C in the vertical direction, to match the levels of sites A, B, C, and D. Maybe extend the horizontal lines also to panel A.Figure 3—figure supplement 2C: The open TRPM2 pore profile (yellow line) is from zebrafish (6drj), whereas the closed profile (blue line) is from a low-resolution human TRPM2 structure (6mix). Maybe replace blue profile with the profile for the apo zebrafish structure (6drk)?

We have made the requested substitution.

23) "white to blue": should be white to orange.

Fixed, thank you.

We have made additional changes in response to feedback on the BioRxiv preprint from other authors in the field, as described below. We thank them for their feedback.

We have updated the discussion of 2-APB mutagenesis experiments to elaborate on the functional studies carried out in Singh, 2018 and Zubcevic, 2019.

A citation was added to the discussion of pi helical segments in S6 (McGoldrick, 2017).

We made further additions to the discussions of ligand binding sites beyond those suggested in point 5 by the reviewers, related to recent papers and suggestions from authors.